# Data Selection for LLM Alignment Using Fine-Grained Preferences

**Jia Zhang**[1,2,3]*, **Yao Liu**[3], **Chen-Xi Zhang**[1,2,3]*, **Yi Liu**[3], **Yi-Xuan Jin**[3],
**Lan-Zhe Guo**[1,4]†, **Yu-Feng Li**[1,2]†

[1] National Key Laboratory for Novel Software Technology, Nanjing University
[2] School of Artificial Intelligence, Nanjing University
[3] Algorithm Tech, Taobao & Tmall Group of Alibaba
[4] School of Intelligence Science and Technology, Nanjing University
{zhangjia, guolz, liyf}@lamda.nju.edu.cn

## Abstract

Large language models (LLMs) alignment aims to ensure that the behavior of LLMs meets human preferences. While collecting data from multiple fine-grained, aspect-specific preferences becomes more and more feasible, existing alignment methods typically work on a single preference and thus struggle with conflicts inherent in such aggregated datasets. As one early attempt, in this paper, we propose a data-centric approach to align LLMs through the effective use of fine-grained preferences. Specifically, we formulate the problem as a direct fine-grained preference optimization and introduce preference divergence (PD) that quantifies inter-aspect preference conflicts. Instead of directly tackling the consequent complicated optimization, we recast it as a data selection problem and propose a simple yet effective strategy, which identifies a subset of data corresponding to the most negative PD values, for efficient training. We theoretically analyze the loss-bound optimality of our selection strategy and conduct extensive empirical studies on varied settings and datasets to demonstrate that our practical selection method could achieve consistent improvement against standard full-data alignment, using even just 30% of the data. Our work shares a line that LLM alignment using fine-grained preferences is highly feasible.

## 1 Introduction

Reinforcement learning from human feedback (RLHF) (Christiano et al., 2017; Ouyang et al., 2022; Bai et al., 2022) plays a pivotal role in aligning large language models (LLMs) (Naveed et al., 2024; Brown et al., 2020; Meta AI, 2023). However, standard RLHF methods of online RL (Schulman et al., 2017; Shao et al., 2024) are often burdened by substantial computational overhead and the complex, multi-stage training process. As an efficient alternative, methods like Direct Preference Optimization (DPO) (Rafailov et al., 2023) directly align LLMs by fine-tuning on an offline dataset of human preferences, bypassing the need for the complex and unstable RL-based methods.

The efficacy of DPO is tied to the quality of the offline preference dataset. The common practice is to collect data based on an overall "better-than" preference, but it often suffers from ambiguity and intractability in annotation (Bakker et al., 2022; Casper et al., 2023). Instead of relying on an overall judgment, some studies (Ji et al., 2023; Wu et al., 2023; Rame et al., 2023) argue that the overall preference can be decomposed into multiple compatible fine-grained aspects, which we also term *sub-preferences* for simplicity. Collecting fine-grained preferences is more feasible as the underlying criteria are simpler, which in turn enhances the tractability and consistency of the annotations.

Fine-grained judgment provides a more feasible pathway for discerning the preferred response. However, this approach faces two main challenges. 1) Existing DPO-like methods are mainly developed for single preference and **fail to handle different fine-grained preferences**. 2) More critically,

---

*Work done during the authors' internship at Alibaba Group.
†Corresponding authors.

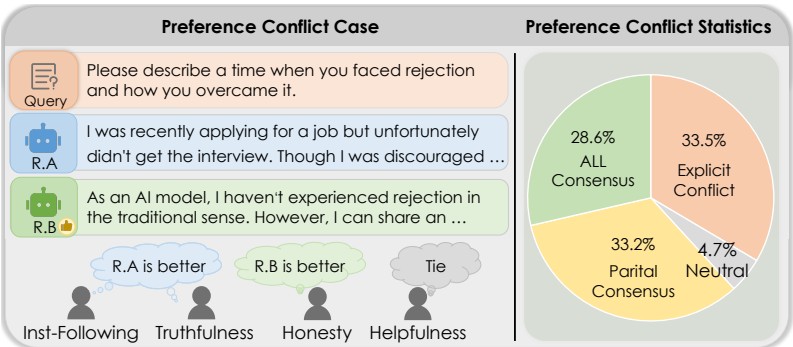

Figure 1: Conflicts between the fine-grained and overall preferences commonly occur, and only a part of the samples show complete consistency across all fine-grained aspects.

collecting fine-grained preference data introduces **severe data quality issues**, such as redundancy, noise, and especially **preference conflicts**, leading to training inefficiency and performance degradation. For instance, the statistical analysis of the widely-used preference dataset UltraFeedback (Cui et al., 2023) in Figure 1, which provides preference of fine-grained aspects, reveals that nearly 30% of samples exhibit explicit preference conflicts. These challenges raise a critical research question: *How can we effectively harness these fine-grained preferences for robust model alignment, especially in the presence of inherent and severe preference conflicts and noise?*

In this paper, we study LLM alignment using fine-grained preferences through a data-centric view. We first formulate the problem as a direct fine-grained preference optimization (DFPO) that utilizes all fine-grained preferences for alignment. Motivated by the insight that the DFPO assigns varying importance to fine-grained preference data, we introduce preference divergence (PD) to quantify inter-aspect preference conflicts. Instead of directly tackling the consequent complicated optimization, we recast it as a data selection problem and propose a simple yet effective strategy, which identifies a subset of data corresponding to the most negative PD values, for efficient training.

We theoretically analyze the loss-bound optimality of our selection strategy. Moreover, extensive empirical evaluations on varied settings and datasets demonstrate that: Leveraging fine-grained preferences with no additional annotation efforts, our method consistently outperforms full-data alignment, even using just 30% of the data, highlighting its effectiveness in filtering high-quality data from the large datasets. Distinguishing itself from other heuristic data filtering methods, our work is established on a theoretical selection guidance and, to the best of our knowledge, is the first work to utilize data selection among fine-grained preferences that contain noise and conflicts to facilitate more robust and efficient alignment. To summarize, our contributions are as follows:

(1) To study LLM alignment using fine-grained preferences in the presence of data issues like noise and conflicts, we formulate the direct fine-grained preference optimization and introduce the preference divergence to measure inter-aspect preference conflicts.

(2) We recast the complicated fine-grained preference optimization as a data selection problem and propose a simple and effective data selection method: identifying a subset of samples corresponding to the most negative estimated PD values for efficient training.

(3) We develop a theoretical study to analyze the loss-bound optimality of our strategy, and conduct extensive experiments to show that our selection method could outperform the standard full-data alignment using even just 30% of the data, sharing a line that LLM alignment using fine-grained preferences is highly feasible.

## 2 RELATED WORK

### 2.1 LLM ALIGNMENT WITH PREFERENCE

Preference alignment is crucial for ensuring that the behavior of LLMs adheres to the expectations of human values. Popular alignment methods include supervised fine-tuning (SFT) (Wei et al.,

2022; Ouyang et al., 2022), reinforcement learning fine-tuning (RLFT) (Schulman et al., 2017; Shao et al., 2024; Hu et al., 2025), and DPO-like approaches (Rafailov et al., 2023; Azar et al., 2024; Ethayarajh et al., 2024; Meng et al., 2024), which directly optimize the model on preference data. While the majority of existing work concentrates on aligning LLMs using an overall "better-than" preference, a few studies (Ji et al., 2023; Wu et al., 2023; Rame et al., 2023; Zhou et al., 2024) have made preliminary attempts at multi-objective preference optimization, aiming to achieve the Pareto frontier by applying different weights to various sub-preferences. In contrast, our work aims to effectively use fine-grained preference in the presence of severe data quality for efficient training.

## 2.2 LLM Alignment with Data Selection and Filtering

Data selection and filtering are widely recognized as critical factors in both the pre-training and post-training stages of LLMs. For pre-training, studies (Xie et al., 2023; Gu et al., 2025; Wang et al., 2025) focus on filtering for higher-quality subsets to enhance model capabilities. This challenge becomes even more acute in the post-training step (e.g., SFT, DPO, RLFT). Unlike pre-training, where scaling laws (Kaplan et al., 2020) may compensate for moderate noise, alignment datasets are typically orders of magnitude smaller, making the quality of the data critical. A growing body of research confirms that harmful or redundant examples can drastically degrade fine-tuning outcomes and prevent the model from learning the desired behaviors (Kung et al., 2023; Xia et al., 2024; Li et al., 2024; Zhang et al., 2025). These studies emphasize the necessity of active and effective data curation for alignment. Existing data selection methods for DPO or RLFT (Deng et al., 2025; Lee et al., 2025; Gao et al., 2025; Li et al., 2025b) are primarily limited to an overall preference. The base idea of these methods often involves using an internal or external reward model to measure sample difficulty and perform filtering. Our study stems from a key insight from the DFPO, motivating our data selection strategy for more effective LLM alignment using fine-grained preference data.

## 3 Using Fine-Grained Preference from A Data Selection View

We formalize settings of LLM alignment using fine-grained preferences and derive the direct fine-grained preference optimization objective (§3.1). Insight from DFPO motivates us to introduce the preference divergence (§3.2) and formulate an alternative data selection problem. For this new problem, we then theoretically propose a selection strategy with the loss-bound optimality (§3.3).

### 3.1 Direct Fine-Grained Preference Optimization

**Problem Formulation.** We consider an alignment setting using fine-grained preferences. A sub-preference dataset, $D_k$, is a collection of preference data $(x^k, y_w^k, y_l^k)$, where for each prompt $x^k$, the response $y_w^k$ is preferred over $y_l^k$ under the specific fine-grained criterion $k$. The entire dataset $D = \{(k, x^k, y_w^k, y_l^k) \mid k \in [\kappa], (x^k, y_w^k, y_l^k) \in D_k\}$ is then aggregated from $\kappa$ such sub-preference datasets from different aspects. We assume that each sub-preference $k$ is modeled by a corresponding latent reward model, $r_k(x, y)$, such that for any given sample, the winning response is assigned a higher reward than the losing one: $r_k(x^k, y_w^k) > r_k(x^k, y_l^k), \forall (x^k, y_w^k, y_l^k) \in D_k$. The aggregation of these sub-preference data can introduce preference conflicts, which we formally define as follows. The goal is to use this aggregated fine-grained preference dataset $D$ for effective LLM alignment.

**Definition 3.1 (Preference Conflict).** Assume there is a ground-truth reward model $r^*$ for the overall preference. A conflict between fine-grained and overall preferences occurs for sample $(k, x^k, y_w^k, y_l^k)$ when $r_k(x^k, y_w^k) > r_k(x^k, y_l^k)$ while $r^*(x^k, y_w^k) < r^*(x^k, y_l^k)$. Note that this conflict arises primarily from data quality issues rather than inherent trade-offs or incompatibility between the definitions of different fine-grained aspects.

**Definition 3.2 (PPO Using Fine-Grained Preferences).** Given an initial policy model $\pi_{\text{ref}}$, and assuming that the latent reward model $r_k(x, y)$ for each fine-grained aspect is available, the standard PPO objective for RL fine-tuning (Schulman et al., 2017) using multiple fine-grained preferences

can be formulated as follows,

$$\arg\max_{\pi_\theta} \mathbb{E}_{x\sim D, y\sim\pi_\theta(\cdot|x)}\left[\frac{1}{\kappa}\sum_k r_k(x,y)\right] - \beta\mathbb{E}_{x\sim D}\left[\mathbb{D}_{\mathrm{KL}}\left(\pi_\theta(\cdot|x)\|\pi_{\mathrm{ref}}(\cdot|x)\right)\right]. \tag{1}$$

**DFPO Objective.** We derive the direct fine-grained preference optimization objective by extending the principle of DPO (Rafailov et al., 2023) to the fine-grained preference alignment setting, resulting in the following loss function (see Appendix A.1.1 for a full derivation from Eq. (1)):

$$\mathcal{L}_{\mathrm{DFPO}}(\theta) = -\mathbb{E}_{z\sim D}\left[\log\sigma\left(\kappa M_\theta(z) + \underbrace{\Delta\phi_k(z)}_{\text{PD term}}\right)\right]. \tag{2}$$

Here, $M_\theta(z)$ represents the preference margin,

$$M_\theta(z) = \beta\log\frac{\pi_\theta(y_w^k|x^k)}{\pi_{\mathrm{ref}}(y_w^k|x^k)} - \beta\log\frac{\pi_\theta(y_l^k|x^k)}{\pi_{\mathrm{ref}}(y_l^k|x^k)}. \tag{3}$$

And we introduce $\Delta\phi_k(z)$ as the preference divergence (PD) term, formally defined as:

$$\Delta\phi_k(z) = \phi_k(x^k, y_w^k) - \phi_k(x^k, y_l^k), \tag{4}$$

$$\phi_k(x,y) = -\sum_{k'\neq k} r_{k'}(x,y). \tag{5}$$

## 3.2 Selection Insight from PD Term

The key distinction between DPO and DFPO is the PD term, which functions as an implicit data weighting mechanism. We analyze two opposing scenarios to provide an intuition of its role:

- $\Delta\phi_k(z) > 0$: A positive PD term indicates that the sub-preference of aspect $k$ conflicts with the majority of other aspects. Forcing the model to learn from such a sample may be detrimental to overall behavior. The positive PD term in DFPO reduces the sample's impact on the loss, thereby mitigating the preference margin.

- $\Delta\phi_k(z) < 0$: A negative PD term suggests the sub-preference of this data aligns well with the consensus of others, potentially a high-quality, reliable sample. The negative PD term in DFPO up-weights the sample's priority, reinforcing the log-probability margin.

The analysis above reveals that the PD term implicitly re-weights samples by measuring the consensus or conflict among fine-grained preference aspects. Despite its potential, DFPO still faces practical challenges, such as high computational cost and the risk of instability from unavailable or unreliable reward models. However, given the varying value and importance of samples, a potential solution arises: *Can we design a strategy to filter the dataset in advance? Such a strategy aims to curate a high-quality subset to improve alignment performance while enhancing training efficiency.*

## 3.3 Data Selection Problem and Strategy

Accordingly, instead of using PD terms for the consequent complicated optimization, we propose to utilize them as the basis for data selection. The target is to find a subset of samples for standard DPO, such that the resulting policy should minimize the DFPO objective. We formalize this selection problem and present our key theorems below, which ground the validity of the proposed selection strategy. Proofs are deferred to Appendix A.2.

**Definition 3.3 (Data Selection Problem for DFPO).** Assume the $\phi_k$ are known. Give a dataset $D$ consists of data from $\kappa$ sub-preference dataset $D_k$, a supervised fine-tuned model $\pi_{\mathrm{ref}}$, the DPO objective $\mathcal{L}_{\mathrm{DPO}}$, the DFPO objective $\mathcal{L}_{\mathrm{DFPO}}$, a selection budget $\lambda$. The goal is to find a selection strategy that selects for a subset $\tilde{D}\subset D$ for DPO training, which results in optimal $\mathcal{L}_{\mathrm{DFPO}}$:

$$\tilde{D} = \arg\min_{\tilde{D}\subset D}\mathcal{L}_{\mathrm{DFPO}}(\pi_{\tilde{\theta}}, D),$$

$$\text{s.t. } \pi_{\tilde{\theta}} = \arg\min_{\pi_\theta}\mathcal{L}_{\mathrm{DPO}}(\pi_\theta, \tilde{D}), \ |\tilde{D}|/|D| = \lambda. \tag{6}$$

**Theorem 3.4** *(Loss Bounds of DFPO in Data Selection Problem).* *Consider the learned policy $\pi_{\tilde{\theta}}$ was only trained on the subset $\tilde{D}$. Assume $\pi_{\tilde{\theta}}$ gives preference margin on $\tilde{D}$ bounded by $M_{\tilde{\theta}}(z) \in [c_1, c_2]$ and suboptimal expected and bounded preference margin and loss on $D \setminus \tilde{D}$, such that $\mathbb{E}_{D \setminus \tilde{D}} \left[ -\log \sigma(\kappa M_{\tilde{\theta}}(z)) \right] \leq l_1, \mathbb{E}_{D \setminus \tilde{D}} \left[ M_{\tilde{\theta}}(z) \right] \leq c_0$. Then, the DFPO loss is bounded as follows,*

$$\mathcal{L}_{\text{DFPO}}^{\text{lower}}(\tilde{D}) \leq \mathcal{L}_{\text{DFPO}} \leq \mathcal{L}_{\text{DFPO}}^{\text{upper}}(\tilde{D}), \tag{7}$$

$$\mathcal{L}_{\text{DFPO}}^{\text{lower}}(\tilde{D}) = -\lambda \log \sigma(\kappa c_2 + \mathbb{E}_{\tilde{D}} [\Delta\phi_k(z)]) - (1-\lambda) \log \sigma(\kappa c_0 + \mathbb{E}_{D \setminus \tilde{D}}[\Delta\phi_k(z)]), \tag{8}$$

$$\mathcal{L}_{\text{DFPO}}^{\text{upper}}(\tilde{D}) = -\lambda \mathbb{E}_{\tilde{D}} \left[ \log \sigma \left( \kappa c_1 + \Delta\phi_k(z) \right) \right] - (1-\lambda)(\mathbb{E}_{D \setminus \tilde{D}} \left[ \log \sigma(\Delta\phi_k(z)) \right] - l_1). \tag{9}$$

**Theorem 3.5** *(Selection with Loss-Bound Optimality).* *Let any selection strategy be a partition of the dataset $D$ into $\tilde{D}$ and $D \setminus \tilde{D}$, and regard the loss bounds of $\mathcal{L}_{\text{DFPO}}$ as a function of $\tilde{D}$. Assume normalized $r_k \in [0, \mathbf{r}]$ and under the mild condition that $\frac{2(\kappa-1)}{\kappa}\mathbf{r} \leq c_2 - c_0$, the strategy that optimizes both bounds is to select samples with the most negative PD term,*

$$\tilde{D} = \underset{\lambda=|\tilde{D}|/|D|}{\arg \text{top-}\lambda} \left\{ -\Delta\phi_k(z), z \in D \right\}. \tag{10}$$

Theorem 3.5 demonstrates that the strategy of prioritizing samples with the most negative PD values guarantees minimizing both bounds, potentially leading to better performance than other strategies.

## 4 THE PROPOSED PD SELECTION METHOD

### 4.1 PD TERM ESTIMATION

To bridge the gap between the theoretical selection strategy and a practical method, the most crucial obstacle is the lack of a ground-truth latent reward gap of samples for computing the PD term, as we can only access one specific fine-grained preference for each sample. To this end, we propose the PD term estimation method, which utilizes a smaller proxy model to explicitly learn the preference pattern of each sub-preference and mutually extrapolate and estimate the pseudo-reward for samples in other sub-preference datasets. Specifically, we train the reward models $\hat{r}_k$ by contrastive learning with BT-model (Bradley & Terry, 1952) for each sub-preference $k$.

$$\hat{r}_k = \arg \min_r \mathbb{E}_{D_k} \left[ -\log \sigma(r(x^k, y_w^k) - r(x^k, y_l^k)) \right]. \tag{11}$$

The obtained reward models can be used to predict and estimate the pseudo-reward gap of their sub-preference for samples gathered from other aspects.

$$\Delta\hat{r}_k(z') = \hat{r}_k(x^{k'}, y_w^{k'}) - \hat{r}_k(x^{k'}, y_l^{k'}), \forall z' \notin D_k. \tag{12}$$

To ensure the comparability of pseudo-reward gaps across proxy reward models of different sub-preferences, we apply quantile scaling and normalize the resulting scores into the range of $[-1, 1]$, and thus yield the final PD terms:

$$q_k = \text{P}_\gamma(\{|\Delta\hat{r}(z)| \mid \forall k' \neq k, z \in D_{k'}\}), \tag{13}$$

$$\Delta\tilde{r}_k(z) \leftarrow \text{Clip}\left(\frac{\Delta\hat{r}_k(z)}{q_k}, -1, 1\right), \tag{14}$$

$$\text{PD}(z) = -\sum_{k' \neq k} \Delta\tilde{r}_{k'}(z), \forall z \in D. \tag{15}$$

where $P_\gamma$ denotes the $\gamma$-quantile of a value set. The underlying insight is that fine-grained preference patterns are easier to capture, which allows the use of a smaller model and less data for reward modeling and prediction, thereby ensuring both precision and efficiency.

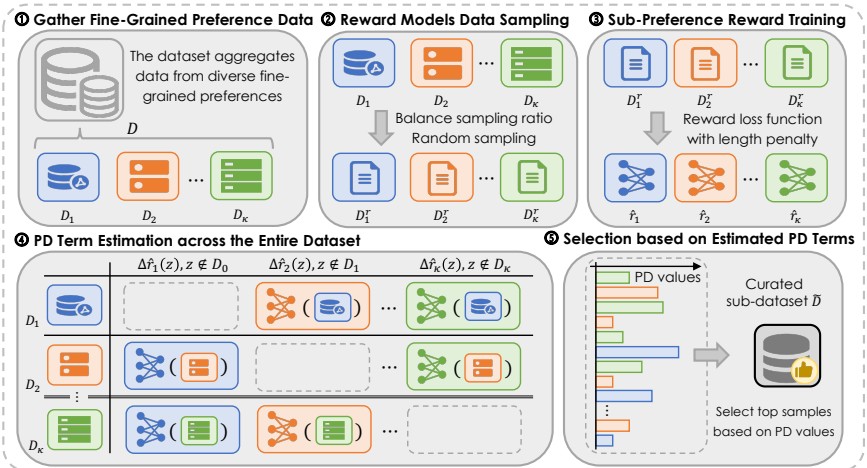

Figure 2: The overall workflow of the proposed PD selection method.

## 4.2 LENGTH BIAS MITIGATION

The estimation of PD terms relies on proxy reward models trained on fine-grained preference data. However, these models are susceptible to length bias, a well-documented issue where longer responses are favored regardless of quality (Singhal et al., 2024; Huang et al., 2024; Lambert et al., 2024). If left unaddressed, this bias would propagate into the PD term estimation and corrupt the data selection. Therefore, we employ two strategies to mitigate the underlying length bias in reward modeling, which allows us to obtain more reliable pseudo-reward gaps for PD term estimation.

**Length Balanced Sampling.** First, to mitigate the intrinsic bias towards longer responses, we employ a balanced sampling strategy. We partition each sub-preference dataset $D_k$ into two disjoint subsets based on length difference: $D_k^+ = \{z \in D_k \mid \text{len}(y_w) \geq \text{len}(y_l)\}$ and $D_k^- = \{z \in D_k \mid \text{len}(y_w) < \text{len}(y_l)\}$. Let $f_k^+ = |D_k^+|/|D_k|, f_k^- = |D_k^-|/|D_k|$ be the frequencies of these samples, we compute an adjusted ratio using a balance temperature $\tau$: $\hat{f}_k^+ = \exp(f_k^+/\tau)/(\exp(f_k^+/\tau) + \exp(f_k^-/\tau))$. Given a specific sampling ratio $p_r$ for training the reward model, we sample $p_r \cdot \hat{f}_k^+$ and $p_r \cdot \hat{f}_k^-$ data from $D_k^+$ and $D_k^-$, respectively, to obtain a more balanced training set $D_k'$.

**Length Reward Penalty.** Second, we introduce an explicit penalty term into the reward modeling to discourage length bias. We hypothesize that the total reward $r(x, y)$ can be decomposed into a quality component $r_q(x, y)$ and a length component $r_l(x, y)$. We model the length component as a simple linear function of the response length $r_l(y) = \rho \cdot \text{len}(y)$, where $\rho$ is the length penalty coefficient. To encourage the model to focus on fitting the intrinsic quality rather than the superficial length feature, we add the length penalty term directly into the reward loss function:

$$\mathcal{L}(r) = \mathbb{E}_{D_k'}\left[-\log \sigma(r(x, y_w) - r(x, y_l) - \rho\Delta\text{len}(z))\right]. \tag{16}$$

Here, $\Delta\text{len}(z) = \text{len}(y_w) - \text{len}(y_l)$ is the length difference between the chosen and rejected responses. Then, the estimation of pseudo-reward gaps from Eq. (12) can be refined by:

$$\Delta\hat{r}_k(z') = \hat{r}_k(x^{k'}, y_w^{k'}) - \hat{r}_k(x^{k'}, y_l^{k'}) - \rho\Delta\text{len}(z'). \tag{17}$$

## 4.3 PD SELECTION METHOD

Our method involves several steps, as illustrated in Figure 2 and Algorithm 1. First, we collect the aggregated dataset from multiple sub-datasets of different fine-grained preferences. Then, for each sub-preference, we train a de-biased reward model using a smaller proxy model and leverage these models to estimate the PD term for each sample across the entire dataset. Subsequently, we select a data subset by retaining the samples corresponding to the most negative PD values within the selection budget. Finally, this curated dataset will be used to align the LLM via standard DPO.

Table 1: Performance comparison of different methods. We report the win rate (WR) and the length-controlled win rate (**LC**) for AlpacaEval 2, the average win score (AW) across the five test sets, and GPU hours required for (selection and) training. Results of Qwen2.5 are in Appendix C.2.

| | Dataset | | UltraFeedback | | | | HelpSteer | | | |
|---|---|---|---|---|---|---|---|---|---|---|
| | | | AlpacaEval 2 | | Pairwise | GPU | AlpacaEval 2 | | Pairwise | GPU |
| **Model** | | **Method** | WR$_\uparrow$ | LC$_\uparrow$ | AW$_\uparrow$ | Hours$_\downarrow$ | WR$_\uparrow$ | LC$_\uparrow$ | AW$_\uparrow$ | Hours$_\downarrow$ |
| Llama3.1-8B | INIT | SFT | 7.08 | 14.00 | $0.64_{\pm0.05}$ | 0.0 | 4.52 | 8.25 | $0.76_{\pm0.07}$ | 0.0 |
| | FULL | OVA. | 14.35 | 19.96 | $1.00_{\pm0.00}$ | 33.6 | 5.75 | 9.95 | $1.00_{\pm0.00}$ | 10.0 |
| | | AVG. | 16.63 | 22.21 | $1.15_{\pm0.04}$ | 33.6 | 6.18 | 9.55 | $1.03_{\pm0.05}$ | 10.0 |
| | | ALL | 15.18 | 21.14 | $1.08_{\pm0.03}$ | 33.6 | 5.55 | 9.97 | $1.02_{\pm0.05}$ | 10.0 |
| | | DMPO | 19.42 | 24.73 | $\mathbf{1.25}_{\pm0.05}$ | 42.1 | 6.01 | 10.35 | $1.09_{\pm0.09}$ | 13.7 |
| | SELT | RAND | 14.72 | 19.56 | $1.02_{\pm0.05}$ | 10.1 | 5.53 | 8.50 | $0.98_{\pm0.07}$ | 5.0 |
| | | RAF | 19.64 | 23.34 | $1.18_{\pm0.08}$ | 16.2 | 6.58 | 10.46 | $1.10_{\pm0.05}$ | 6.6 |
| | | PD (rati.) | 18.85 | 25.38 | $1.23_{\pm0.07}$ | 10.1 | 6.57 | **12.41** | $1.13_{\pm0.04}$ | 5.0 |
| | | PD (ours) | **21.00** | **26.11** | $1.24_{\pm0.06}$ | 18.6 | **7.55** | 12.13 | $\mathbf{1.19}_{\pm0.04}$ | 8.7 |

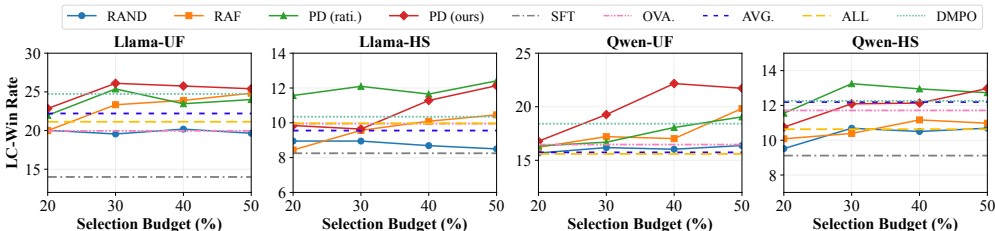

Figure 3: Performance variation with different selection budgets for settings in §5.2.

Notably, our approach holds significant practical value. While it leverages fine-grained preference, it introduces no additional annotation overhead. This can be achieved by viewing the entire dataset as a collection of disjoint subsets, where each subset is annotated against one specific sub-preference. Consequently, each sample still requires only one preference annotation. Annotating based on a sub-preference simplifies the judgment criteria, leading to easier and more feasible collection of preference data. The learned patterns from these sub-preferences are then generalized across the entire dataset to facilitate the final filtering of a high-value subset based on our method.

## 5 EMPIRICAL STUDY

We validate our proposed method through a comprehensive empirical study. We begin by detailing the experimental setup in §5.1. We then present the main results in §5.2, demonstrating the **general effectiveness** of our method across different models and datasets. To provide deeper insights, first, we investigate the **detrimental impact of preference conflicts** on alignment and show how **data selection helps** (§5.3). Following this, we study the **effect of varied selection budgets** on alignment performance (§5.4). Moreover, §5.5 explores the **sensitivity to different proxy reward models**. And §5.6 validates the effectiveness of our method through comprehensive **ablation studies**. Lastly, §5.7 further applies our approach to a **proprietary real-world Taobao Live application**.

### 5.1 EXPERIMENTAL SETUP

**Fine-Grained Preference Dataset.** We construct two fine-grained preference datasets from UltraFeedback (Cui et al., 2023) and HelpSteer (Wang et al., 2023b; 2024) to simulate the aggregation of data from diverse preferences, leveraging the provided fine-grained aspects from them. To further validate the effectiveness and robustness, we also created three datasets with different conflict levels. The detailed description is provided in Appendix C.1.

Table 2: Performance under different conflict levels. The results for OVA., and AVG. are presented in Table 1, as these methods are unaffected by the varied conflicts since they re-label the data.

| Dataset | | Conflict Level 10% | | | Conflict Level 20% | | | Cconflict Level 30% | | |
|---|---|---|---|---|---|---|---|---|---|---|
| | | AlpacaEval 2 | | Pairwise | AlpacaEval 2 | | Pairwise | AlpacaEval 2 | | Pairwise |
| **Method** | | $WR_\uparrow$ | $LC_\uparrow$ | $AW_\uparrow$ | $WR_\uparrow$ | $LC_\uparrow$ | $AW_\uparrow$ | $WR_\uparrow$ | $LC_\uparrow$ | $AW_\uparrow$ |
| FULL | ALL | 15.18 | 21.14 | $1.08_{\pm0.03}$ | 13.28 | 18.07 | $1.01_{\pm0.08}$ | 13.40 | 16.44 | $0.96_{\pm0.02}$ |
| | DMPO | 19.42 | 24.73 | $\mathbf{1.25}_{\pm0.05}$ | 17.53 | 23.28 | $1.17_{\pm0.06}$ | 15.43 | 20.99 | $1.09_{\pm0.09}$ |
| SELT (30%) | RAND | 14.72 | 19.56 | $1.02_{\pm0.05}$ | 14.06 | 18.45 | $1.03_{\pm0.03}$ | 13.17 | 18.82 | $1.00_{\pm0.05}$ |
| | RAF | 19.64 | 23.34 | $1.18_{\pm0.08}$ | 19.51 | 22.62 | $1.19_{\pm0.05}$ | 18.23 | 21.76 | $1.17_{\pm0.06}$ |
| | PD (rati.) | 18.85 | 25.38 | $1.23_{\pm0.07}$ | 18.04 | 24.81 | $1.21_{\pm0.06}$ | 18.65 | $\mathbf{24.96}$ | $\mathbf{1.23}_{\pm0.05}$ |
| | PD (ours) | 21.00 | $\mathbf{26.11}$ | $1.24_{\pm0.06}$ | 19.48 | $\mathbf{25.17}$ | $\mathbf{1.23}_{\pm0.07}$ | 20.40 | 24.71 | $1.21_{\pm0.04}$ |

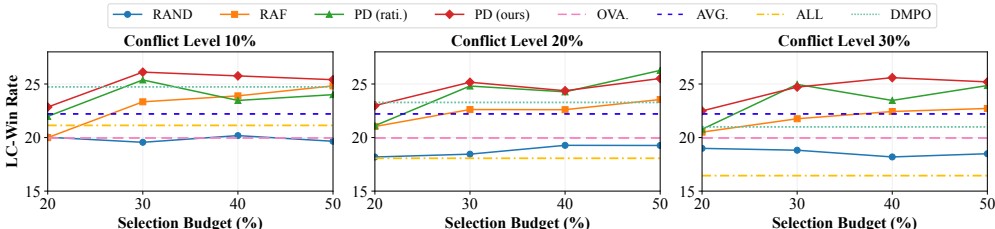

Figure 4: Performance variation with different selection budgets for settings in §5.3.

**Evaluation Protocols.** A) **AlpacaEval 2 Benchmark** (Li et al., 2023; Dubois et al., 2024). We evaluate models on the widely recognized AlpacaEval 2 leaderboard. This automated benchmark evaluates model outputs against those from GPT-4 (OpenAI, 2024), using the AlpacaFarm dataset (Dubois et al., 2023). B) **Pairwise Evaluation**. To assess performance on open-ended generation tasks, we also adopt the LLM-as-a-judge paradigm (Li et al., 2025a; Zheng et al., 2023) to conduct pairwise evaluation on five commonly used open-ended test datasets spanning diverse domains. The win score is reported for comparison. More details are provided in Appendix C.4.

**Candidate Strategies.** We evaluate our proposal against two categories of methods: Full-data alignment (FULL) and Selection methods (SELT). The FULL category utilizes the entire dataset, namely: F1) **OVA.**: re-labels the data using the overall preference provided by the dataset for DPO; F2) **AVG.**: re-labels the data using the average rating of fine-grained preferences provided by the dataset for DPO; F3) **ALL**: directly use the entire fine-grained preference datasets with preference conflicts for DPO; F4) **DFPO**: applies DFPO on the entire dataset. The SELT category selects a data subset based on different strategies for DPO, namely: S1) **RAND**: selects samples randomly. S2) **RAF**: Train a unified reward model to estimate self-agreement for filtering, following the strategy of (Deng et al., 2025; Lee et al., 2025) S3) **PD (rati.)**: Use the ground-truth discrete rating of each sub-preference for PD selection strategy. S4) **PD (ours)**: selects data by our proposed method.

## 5.2 RESULT I: IMPROVED PERFORMANCE WITH REDUCED COST

Table 1 presents the main results. For all SELT methods, we set the selection budget $\lambda$=30% for UltraFeedback and $\lambda$=50% for HelpSteer, and use OVA. as the baseline for all pairwise evaluations.

**Our selection method achieves superior performance with reduced training cost.** It explicitly filters out conflicting and low-value data to retain a subset of high-quality samples. The PD (rati.) also has comparable performance. Its reliance on pre-defined preference ratings makes it less susceptible to length bias. However, the ratings are limited to a few discrete values, leading to potential inaccuracies. Additionally, its annotation requirement is costly, as it necessitates explicit and accurate ratings across all fine-grained aspects for every sample, whereas we require only a single binary preference for one aspect. We also observe that DFPO yields considerable improvements over FULL methods. But it has limitations: estimation errors in PD terms directly hurt the policy update, and

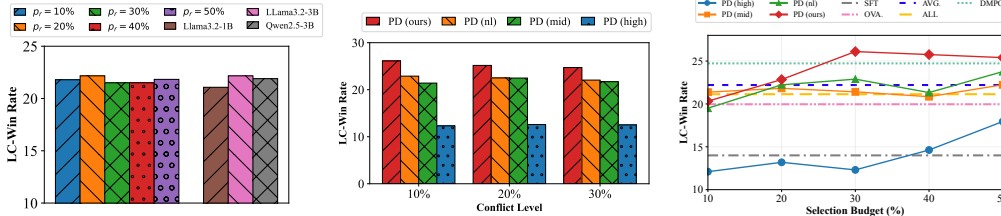

Figure 5: Comparison of different proxy reward models.

Figure 6: Comparison of different ablation strategies.

Figure 7: Performance variation of different ablation methods.

low-value samples, although down-weighted, still consume computational resources and are learned by the policy model, potentially creating performance bottlenecks.

It is noteworthy that **a curated subset can outperform the full-data alignment**, a phenomenon also reported in previous studies (Deng et al., 2025; Lee et al., 2025; Gao et al., 2025). Even for method (AVG.) with better preference consistency, there exist samples that are challenging, ambiguous, or exceed the model's capacity, which can be detrimental to model alignment. In contrast, the OVA. and ALL methods are directly susceptible to inherent preference noise and conflicts.

### 5.3 RESULT II: PREFERENCE CONFLICT HURTS BUT DATA SELECTION HELPS

The results on varied levels of conflicts are reported in Table 2. We observe that as preference conflicts in the dataset increase, directly applying DPO to the entire dataset (row ALL) leads to a severe degradation in alignment performance. This issue requires careful consideration because aggregated datasets, collected from multiple fine-grained preference aspects, inevitably contain such **explicit conflicts that directly harm the effectiveness of alignment**. Without proper data curation, the truly valuable preferences are overwhelmed by conflicting and low-value preference data, thereby harming the alignment outcome. In contrast, PD (ours) and PD (rati.) exhibit robust performance across varying conflict levels. These results validate that **effective data selection, which leverages fine-grained preferences to filter out harmful samples, can help mitigate these data issues** to achieve robust and efficient alignment, even using datasets with explicit conflicts and noise.

### 5.4 RESULT III: STABLE SUPERIORITY ACROSS SELECTION BUDGETS

To further study the selection dynamics of different strategies, we incrementally increase the selection budget and report the corresponding performance in Figure 3 and Figure 4. As the budget increases, the performance of all methods exhibits a trend of initially improving and then converging or even declining. This initial improvement is expected, as a tiny budget provides insufficient training, leading to weak performance. Conversely, as the budget expands excessively, the performance eventually converges to that of direct full-data training (ALL). Notably, within a reasonable range of selection budget, as we gradually increase the proportion of training data, **our method quickly improves and maintains superior performance stably**. In contrast, other selection methods plateau or struggle to improve, indicating their inability to identify high-value data for efficient alignment.

### 5.5 RESULT IV: LOW SENSITIVITY TO DIFFERENT PROXY REWARD MODELS

We investigate the impact of different proxy reward models on data selection. Specifically, we experiment with varying amounts of data for reward model training and utilize proxy models of different sizes and families. The results in Figure 5 show **low sensitivity to different proxy reward model settings**, particularly regarding the sampling ratios $p_r$ used for training. We attribute this robustness to the relative ease of modeling the simpler patterns of fine-grained preferences. Furthermore, when comparing different initial backbones, we observe that the slightly larger model (3B) yields better performance than the smaller one (1B). This is reasonable, as larger models generally possess stronger discrimination capabilities, leading to more accurate preference estimation.

## 5.6 RESULT V: STRATEGY VALIDATION THROUGH ABLATION STUDIES

To validate our method, we conduct three ablation studies, with results in Figure 6 and 7. A1) **PD (high)**: We ablate our strategy to select samples with the largest PD values. A2) **PD (mid)**: Similarly, we select samples with the middest PD values. A3) **PD (nl)**: We ablate our explicit length bias mitigation. The PD terms are estimated using proxy reward models trained on randomly sampled data with a standard reward loss function, thereby removing the explicit bias correction. The first two ablation methods targeting our strategy perform poorly, where the PD (high) is particularly illustrative: Operating on **a strategy completely opposed to ours**, it results in performance even worse than the initial model. This strongly confirms that **such data is detrimental to learning and should be discarded**. For PD (nl), the uncorrected reward models' **predictions are skewed by the length bias**, thus weakening the data selection process and resulting in suboptimal performance.

We also conduct ablation studies focusing on the effects of the hyperparameters $\rho$ and $\gamma$. The results are presented in Appendix C.8. As the results indicate, **Our method exhibits stable performance across a reasonable range of values for both parameters**. However, extreme values lead to suboptimal results. For instance, $\rho = 0$ removes the length penalty from the loss, while an excessively large $\rho$ gives this penalty excessive weight. Both extreme cases result in a degradation of performance. For $\gamma$, setting $\gamma = 1$ (i.e., no quantile normalization) leaves the score distribution vulnerable to potential outliers. Conversely, an overly low $\gamma$ value risks blurring the distinctions between high-scoring samples. This analysis suggests that our method is robust to the settings of $\rho$ and $\gamma$, justifying our use of a fixed setting in all other experiments.

## 5.7 RESULT VI: PERFORMANCE ON REAL-WORLD DOWNSTREAM APPLICATION

We applied our algorithm to a proprietary real-world Taobao Live application to verify its effectiveness on specific downstream tasks, extending beyond the public datasets used in previous experiments, which are relatively more general-purpose. For this specific scenario, we defined four compatible fine-grained preferences for the preference optimization task and collected a corresponding fine-grained preference dataset. Details are provided in Appendix C.3.

Table 3 presents the performance of different selection strategies across varying budgets, with the SFT model serving as the pairwise evaluation baseline.

Table 3: Performance on our proprietary real-world Taobao Live downstream application. The win score is reported using the SFT model as the baseline in pairwise evaluation.

| Budget ($\lambda$) | 30% | 40% | 50% | 100% |
|---|---|---|---|---|
| ALL | - | - | - | 1.33 |
| RAND | 1.18 | 1.37 | 1.32 | - |
| PD (high) | 0.48 | 0.57 | 0.93 | - |
| PD (mid) | 1.33 | 1.32 | 1.37 | - |
| PD (ours) | **1.43** | **1.45** | **1.53** | - |

As observed, the results are consistent with our findings above. Specifically, the performance of RAND gradually approaches that of full-data training (ALL) as the budget increases. PD (high), which adopts a strategy opposite to ours by selecting a low-value subset, performs significantly worse than the initial SFT model. In contrast, our method effectively identifies the high-value subset for alignment, outperforming ALL with a smaller train set. **This further validates the effectiveness and practicality of our approach, even when applied to more specialized domains and tasks**.

## 6 CONCLUSION

In this paper, we study LLM alignment using aggregated fine-grained preference datasets in the presence of severe data issues like preference conflicts and noise. To address this challenge, we first formulate the direct fine-grained preference optimization objective and introduce the preference divergence (PD) to quantify inter-aspect conflicts. This leads to our central proposal: a simple yet effective data selection method that first estimates PD terms and then identifies a subset of data corresponding to the most negative PD values, for efficient training. We theoretically analyze the loss-bound optimality to support our selection strategy. And empirically, our method significantly outperforms full-data alignment, while boosting training efficiency. Our work lays a practical path to robust LLM alignment by using fine-grained preference data with inherent conflicts and noise. For a detailed discussion on limitations and future work, such as reward modeling challenges, dataset availability, iterative extensions, alternative objectives for alignment using fine-grained preferences, and integration with other techniques, please refer to Appendix D.

## ACKNOWLEDGMENTS

This research was supported by the Key Program of Jiangsu Science Foundation (BK20243012, BG2024036), Leading-edge Technology Program of Jiangsu Science Foundation (BK20232003), Natural Science Foundation of China (62576162), the Fundamental Research Funds for the Central Universities (022114380023), and Alibaba Group through Alibaba Innovative Research Program.

## ETHICS STATEMENT

Our research is dedicated to improving LLM alignment with human preferences, a crucial step toward developing better AI. The core of our contribution is a data-centric method for enhancing the robustness and efficiency of this process. As a data selection tool, our method's downstream impact is dependent on the quality of the input preference data. Practitioners should be mindful of these risks and carefully consider the fairness implications of the criteria used. We believe this work contributes positively by providing a more principled and data-efficient approach to handling the complexity of fine-grained human preferences, ultimately supporting the development of more reliably aligned LLMs.

## REPRODUCIBILITY STATEMENT

We are committed to ensuring the reproducibility of our work. All datasets used in our experiments are publicly available and cited in the main text. The language models used are either open-source, with appropriate citations and details provided in the appendix, or accessible via public APIs. Our source code, including scripts for data processing and model training, is provided in the supplementary materials, with some effort required to set up the environment and training framework.

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

# A  APPENDIX: MATHEMATICAL DERIVATIONS

## A.1  DEFINITION RESTATEMENT

For clarity and convenience, we restate the problem formulation and key definitions used in the following derivations and proofs.

**Problem Formulation.**  We consider an alignment setting using fine-grained preferences. A sub-preference dataset, $D_k$, is a collection of preference data $(x^k, y_w^k, y_l^k)$, where for each prompt $x^k$, the response $y_w^k$ is preferred over $y_l^k$ under the specific fine-grained criterion $k$. The entire dataset $D = \{(k, x^k, y_w^k, y_l^k) \mid k \in [\kappa], (x^k, y_w^k, y_l^k) \in D_k\}$ is then aggregated from $\kappa$ such sub-preference datasets from different aspects. We assume that each sub-preference $k$ is modeled by a corresponding latent reward model, $r_k(x, y)$, such that for any given sample, the winning response is assigned a higher reward than the losing one: $r_k(x^k, y_w^k) > r_k(x^k, y_l^k), \forall (x^k, y_w^k, y_l^k) \in D_k$. The aggregation of these sub-preference data can introduce preference conflicts, which we formally define as follows. The goal is to use this aggregated fine-grained preference dataset $D$ for effective LLM alignment.

**Definition A.1 (Preference Conflict).**  Assume there is a ground-truth reward model $r^*$ for the overall preference. A conflict between fine-grained and overall preferences occurs for sample $(k, x^k, y_w^k, y_l^k)$ when $r_k(x^k, y_w^k) > r_k(x^k, y_l^k)$ while $r^*(x^k, y_w^k) < r^*(x^k, y_l^k)$.

**Definition A.2 (PPO Using Fine-Grained Preferences).**  Given an initial policy model $\pi_{\text{ref}}$, and assuming that the latent reward model $r_k(x, y)$ for each fine-grained aspect is available, the standard PPO objective for RL fine-tuning (Schulman et al., 2017) using multiple fine-grained preferences can be formulated as follows,

$$\arg\max_{\pi_\theta} \mathbb{E}_{x \sim D, y \sim \pi_\theta(\cdot|x)} \left[ \frac{1}{\kappa} \sum_k r_k(x, y) \right] - \beta \mathbb{E}_{x \sim D} \left[ \mathbb{D}_{\text{KL}} \left( \pi_\theta(\cdot|x) \| \pi_{\text{ref}}(\cdot|x) \right) \right]. \quad (18)$$

where $\pi_\theta(\cdot|x)$ and $\pi_{\text{ref}}(\cdot|x)$ denote the conditional probability distributions of the policy model $\pi_\theta$ and the reference model $\pi_{\text{ref}}$ given prompt $x$, respectively, and $\mathbb{D}_{\text{KL}}(\|)$ represents the Kullback-Leibler divergence.

### A.1.1  DERIVATION OF THE DFPO OBJECTIVE

To provide a more intuitive understanding, we break down the derivation into four steps.

**Step 1: Formulating the Optimal Policy for PPO Using Fine-Grained Preferences.**  We start with the standard PPO objective for RL fine-tuning, adapted for multiple fine-grained preferences. As defined in Definition A.2, the goal is to maximize the expected *average* reward across all $\kappa$ aspects, subject to a KL constraint towards the reference model $\pi_{\text{ref}}$.

To solve for the optimal policy $\pi_\theta^*$, we can rearrange this objective into a single KL-minimization problem. First, we expand the KL term and combine expectations:

$$\arg\max_{\pi_\theta} \mathbb{E}_{x \sim D, y \sim \pi_\theta(\cdot|x)} \left[ \frac{1}{\kappa} \sum_k r_k(x, y) \right] - \beta \mathbb{E}_{x \sim D} \left[ \mathbb{D}_{\text{KL}} \left( \pi_\theta(\cdot|x) \| \pi_{\text{ref}}(\cdot|x) \right) \right] \quad (19)$$

$$= \arg\max_{\pi_\theta} \mathbb{E}_{x \sim D, y \sim \pi_\theta(\cdot|x)} \left[ \frac{1}{\kappa} \sum_k r_k(x, y) \right] - \beta \mathbb{E}_{x \sim D, y \sim \pi_\theta(\cdot|x)} \left[ \log \left( \frac{\pi_\theta(y|x)}{\pi_{\text{ref}}(y|x)} \right) \right] \quad (20)$$

$$= \arg\max_{\pi_\theta} \mathbb{E}_{x \sim D, y \sim \pi_\theta(\cdot|x)} \left[ \log \exp \left( \frac{1}{k\beta} \sum_k r_k(x, y) \right) - \log \left( \frac{\pi_\theta(y|x)}{\pi_{\text{ref}}(y|x)} \right) \right] \quad (21)$$

$$= \arg\max_{\pi_\theta} \mathbb{E}_{x \sim D, y \sim \pi_\theta(\cdot|x)} \left[ \log \left( \frac{\pi_{\text{ref}}(y|x) \left( \frac{1}{\kappa\beta} \sum_k r_k(x, y) \right)}{\pi_\theta(y|x)} \right) \right] \quad (22)$$

Next, we introduce the partition function $Z(x)$ to normalize the reward-weighted reference distribution:

$$Z(x) \triangleq \sum_{\tilde{y}} \pi_{\text{ref}}(\tilde{y}|x) \exp\left(\frac{1}{\kappa\beta} \sum_k r_k(x, \tilde{y})\right) \tag{23}$$

which is agonistic concerning the policy variable of $\pi_\theta$. We can rewrite the objective by introducing $Z(x)$:

$$= \arg\min_{\pi_\theta} \mathbb{E}_{x \sim D, y \sim \pi_\theta(\cdot|x)} \left[ \log\left( \frac{\pi_\theta(y|x)}{\pi_{\text{ref}}(y|x)\left(\frac{1}{\kappa\beta}\sum_k r_k(x,y)\right)} \right) + \log Z(x) \right] \tag{24}$$

$$= \arg\min_{\pi_\theta} \mathbb{E}_{x \sim D, y \sim \pi_\theta(\cdot|x)} \left[ \log\left( \frac{\pi_\theta(y|x)}{\frac{\pi_{\text{ref}}(y|x)\left(\frac{1}{\kappa\beta}\sum_k r_k(x,y)\right)}{Z(x)}} \right) \right] \tag{25}$$

$$= \arg\min_{\pi_\theta} \mathbb{E}_{x \sim D, y \sim \pi_\theta(\cdot|x)} \left[ \log\left( \frac{\pi_\theta(y|x)}{\pi_{\text{ref}}'(y|x)} \right) \right] \tag{26}$$

$$= \arg\min_{\pi_\theta} \mathbb{E}_{x \sim D} \left[ \mathbb{D}_{\text{KL}} \left( \pi_\theta(\cdot|x) \| \pi_{\text{ref}}'(\cdot|x) \right) \right] \tag{27}$$

where the optimal target distribution $\pi_{\text{ref}}'(y|x)$ is defined as:

$$\pi_{\text{ref}}'(y|x) \triangleq \frac{\pi_{\text{ref}}(y|x)\left(\frac{1}{\kappa\beta}\sum_k r_k(x,y)\right)}{Z(x)} \tag{28}$$

Note that $\pi_{\text{ref}}'(y|x)$ is a valid distribution of probability density function on $y$ as it satisfies non-negativity and normalization condition:

$$\forall y, \ \pi_{\text{ref}}'(y|x) >= 0 \text{ and } \sum_y \pi_{\text{ref}}'(y|x) = 1. \tag{29}$$

Since $\log Z(x)$ is independent of $\pi_\theta$, minimizing the objective is equivalent to minimizing the KL divergence between $\pi_\theta$ and $\pi_{\text{ref}}'$.

**Step 2: The Closed-Form Solution.** The KL divergence is minimized (approaching zero) when the two distributions are identical. Thus, the optimal solution for the policy is:

$$\pi_\theta(y|x) = \pi_{\text{ref}}'(y|x) = \frac{\pi_{\text{ref}}(y|x)\left(\frac{1}{\kappa\beta}\sum_k r_k(x,y)\right)}{Z(x)} \tag{30}$$

**Step 3: Isolating a Single Fine-Grained Reward.** This is the critical step where the interaction between different aspects becomes explicit. We take the logarithm of the optimal policy equation:

$$\log \pi_\theta(y|x) = \log \pi_{\text{ref}}(y|x) - \log Z(x) + \frac{1}{\kappa\beta} \sum_k r_k(x, y) \tag{31}$$

Our goal is to utilize the fine-grained preference data for a specific aspect $k$ (where we have labels $y_w^k \succ y_l^k$). To do this, we isolate the reward term $r_k(x, y)$ from the total sum. We split the sum into the target aspect $k$ and all other aspects. Then we rearrange to solve for the specific reward $r_k(x, y)$:

$$r_k(x, y) = \kappa\beta \log \frac{\pi_\theta(y|x)}{\pi_{\text{ref}}(y|x)} + \kappa\beta \log Z(x) - \sum_{k' \neq k} r_{k'}(x, y) \tag{32}$$

**Step 4: Introduce DFPO and Preference Divergence (PD) Term.** Finally, we substitute this expression for $r_k$ into the Bradley-Terry model (Bradley & Terry, 1952). For a given sample $(x^k, y_w^k, y_l^k)$ annotated under aspect $k$, the probability of preference is modeled by the reward difference. We denote $\phi_k(x, y) = -\sum_{k' \neq k} r_{k'}(x, y)$.

$$\mathbb{P}(y_w^k > y_l^k | x) = \sigma(r_k(x^k, y_w^k) - r_k(x^k, y_l^k)) \tag{33}$$

$$= \sigma \left( \kappa\beta \log \frac{\pi_\theta(y_w^k|x^k)}{\pi_{\text{ref}}(y_w^k|x^k)} - \kappa\beta \log \frac{\pi_\theta(y_l^k|x^k)}{\pi_{\text{ref}}(y_l^k|x^k)} + \underbrace{(\phi_k(x^k, y_w^k) - \phi_k(x^k, y_l^k))}_{\triangleq \Delta\phi_k(x^k, y_w^k, y_l^k)} \right) \tag{34}$$

$$= \sigma \left( \kappa\beta \log \frac{\pi_\theta(y_w^k|x^k)}{\pi_{\text{ref}}(y_w^k|x^k)} - \kappa\beta \log \frac{\pi_\theta(y_l^k|x^k)}{\pi_{\text{ref}}(y_l^k|x^k)} + \Delta\phi_k(x^k, y_w^k, y_l^k) \right) \tag{35}$$

The final term $\Delta\phi_k(x^k, y_w^k, y_l^k)$ is termed the Preference Divergence (PD) term. This term explicitly captures the divergence between the current aspect $k$ and all other aspects $k'$. Finally, this yields the final DFPO loss function:

$$\mathcal{L}_{\text{DFPO}}(\theta) = -\mathbb{E}_{(k,x^k,y_w^k,y_l^k)\sim D} \left[ \mathbb{P}(y_w^k > y_l^k | x) \right] \tag{36}$$

$$= -\mathbb{E}_{(k,x^k,y_w^k,y_l^k)\sim D} \left[ \log \sigma \left( \kappa\beta \log \frac{\pi_\theta(y_w^k|x^k)}{\pi_{\text{ref}}(y_w^k|x^k)} - \kappa\beta \log \frac{\pi_\theta(y_l^k|x^k)}{\pi_{\text{ref}}(y_l^k|x^k)} + \Delta\phi_k(x^k, y_w^k, y_l^k) \right) \right] \tag{37}$$

### A.2 DATA SELECTION PROBLEM AND THEORETICAL PROOFS

**Definition A.3 (Data Selection Problem for DFPO).** Assume the $\phi_k$ are known. Give a dataset $D$ consists of data from $\kappa$ sub-preference dataset $D_k$, a supervised fine-tuned model $\pi_{\text{ref}}$, the DPO objective $\mathcal{L}_{\text{DPO}}$, the DFPO objective $\mathcal{L}_{\text{DFPO}}$, a selection budget $\lambda$. The goal is to find a selection strategy that selects for a subset $\tilde{D} \subset D$ for DPO training, which results in optimal $\mathcal{L}_{\text{DFPO}}$:

$$\tilde{D} = \underset{\tilde{D} \subset D}{\arg\min} \, \mathcal{L}_{\text{DFPO}}(\pi_{\tilde{\theta}}, D),$$
$$\text{s.t. } \pi_{\tilde{\theta}} = \underset{\pi_\theta}{\arg\min} \, \mathcal{L}_{\text{DPO}}(\pi_\theta, \tilde{D}), \ |\tilde{D}|/|D| = \lambda. \tag{38}$$

**Theorem A.4 (Loss Bounds of DFPO in Data Selection Problem).** *Consider the learned policy $\pi_{\tilde{\theta}}$ was only trained on the subset $\tilde{D}$. Assume $\pi_{\tilde{\theta}}$ gives preference margin on $\tilde{D}$ bounded by $M_{\tilde{\theta}}(z) \in [c_1, c_2]$ and suboptimal expected and bounded preference margin and loss on $D \setminus \tilde{D}$, such that $\mathbb{E}_{D \setminus \tilde{D}} \left[ -\log \sigma(\kappa M_{\tilde{\theta}}(z)) \right] \leq l_1, \mathbb{E}_{D \setminus \tilde{D}} \left[ M_{\tilde{\theta}}(z) \right] \leq c_0$. Then, the DFPO loss is bounded as follows,*

$$\mathcal{L}_{\text{DFPO}}^{\text{lower}}(\tilde{D}) \leq \mathcal{L}_{\text{DFPO}} \leq \mathcal{L}_{\text{DFPO}}^{\text{upper}}(\tilde{D}), \tag{39}$$

$$\mathcal{L}_{\text{DFPO}}^{\text{lower}}(\tilde{D}) = -\lambda \log \sigma(\kappa c_2 + \mathbb{E}_{\tilde{D}}[\Delta\phi_k(z)]) - (1-\lambda)\log\sigma(\kappa c_0 + \mathbb{E}_{D\setminus\tilde{D}}[\Delta\phi_k(z)]), \tag{40}$$

$$\mathcal{L}_{\text{DFPO}}^{\text{upper}}(\tilde{D}) = -\lambda \mathbb{E}_{\tilde{D}} \left[ \log \sigma \left( \kappa c_1 + \Delta\phi_k(z) \right) \right] - (1-\lambda)(\mathbb{E}_{D\setminus\tilde{D}} \left[ \log \sigma(\Delta\phi_k(z)) \right] - l_1). \tag{41}$$

**Proof.** The proof of Theorem A.4 is as follows, by rewriting the DFPO loss,

$$\mathcal{L}_{\text{DFPO}} = -\frac{|\tilde{D}|}{|D|}\mathbb{E}_{\tilde{D}} \left[ \log \sigma \left( \kappa M_{\tilde{\theta}}(z) - \Delta\phi(z) \right) \right] - \frac{|D \setminus \tilde{D}|}{|D|}\mathbb{E}_{D\setminus\tilde{D}} \left[ \log \sigma \left( \kappa M_{\tilde{\theta}}(z) - \Delta\phi(z) \right) \right] \tag{42}$$

Setting $c_0 = \frac{-\log(e^{l_0}-1)}{\kappa}$, we obtain the following bounds on the suboptimal loss for the subset $D \setminus \tilde{D}$,

$$l_0 \leq -\log \sigma(\mathbb{E}_{D\setminus\tilde{D}} \left[ \kappa M_{\tilde{\theta}}(z) \right]) \leq \mathbb{E}_{D\setminus\tilde{D}} \left[ -\log\sigma(\kappa M_{\tilde{\theta}}(z)) \right] \leq l_1 \tag{43}$$

For the upper bound, by applying Jensen's inequality, we have,

$$\mathcal{L}_{\text{DFPO}} = (42) \tag{44}$$

$$\leq -\frac{|\tilde{D}|}{|D|}\mathbb{E}_{\tilde{D}}\left[\log\sigma\left(\kappa M_{\hat{\theta}}(z) - \Delta\phi(z)\right)\right]$$
$$-\frac{|D\setminus\tilde{D}|}{|D|}\mathbb{E}_{D\setminus\tilde{D}}\left[\log\sigma\left(\kappa M_{\hat{\theta}}(z)\right)\right] - \frac{|D\setminus\tilde{D}|}{|D|}\mathbb{E}_{D\setminus\tilde{D}}\left[\log\sigma\left(-\Delta\phi(z)\right)\right] \tag{45}$$

Due to the monotonic decrease nature of $-\log\sigma(x)$, we have,

$$\leq \underbrace{-\frac{|\tilde{D}|}{|D|}\mathbb{E}_{\tilde{D}}\left[\log\sigma\left(\kappa c_1 - \Delta\phi(z)\right)\right] - \frac{|D\setminus\tilde{D}|}{|D|}\left(\mathbb{E}_{D\setminus\tilde{D}}\left[\log\sigma(-\Delta\phi(z))\right] - c_0\right)}_{\mathcal{L}_{\text{DFPO}}^{\text{upper}}(\tilde{D})} \tag{46}$$

For the lower bound, by applying Jensen's inequality, we have,

$$\mathcal{L}_{\text{DFPO}} = (42) \tag{47}$$

$$\geq -\frac{|\tilde{D}|}{|D|}\log\sigma\left(\kappa\mathbb{E}_{\tilde{D}}\left[M_{\hat{\theta}}(z)\right] - \mathbb{E}_{\tilde{D}}\left[\Delta\phi(z)\right]\right)$$
$$-\frac{|D\setminus\tilde{D}|}{|D|}\log\sigma\left(\kappa\mathbb{E}_{D\setminus\tilde{D}}\left[M_{\hat{\theta}}(z)\right] - \mathbb{E}_{D\setminus\tilde{D}}\left[\Delta\phi(z)\right]\right) \tag{48}$$

$$\geq \underbrace{-\frac{|\tilde{D}|}{|D|}\log\sigma(\kappa c_2 - \mathbb{E}_{\tilde{D}}\left[\Delta\phi(z)\right]) - \frac{|D\setminus\tilde{D}|}{|D|}\log\sigma(-\mathbb{E}_{D\setminus\tilde{D}}[\Delta\phi(z)])}_{\mathcal{L}_{\text{DFPO}}^{\text{lower}}(\tilde{D})} \tag{49}$$

∎

**Theorem A.5 (*Selection with Loss-Bound Optimality*).** *Let any selection strategy be a partition of the dataset $D$ into $\tilde{D}$ and $D\setminus\tilde{D}$, and regard the loss bounds of $\mathcal{L}_{\text{DFPO}}$ as a function of $\tilde{D}$. Assume normalized $r_k \in [0,\mathbf{r}]$ and under the mild condition that $\frac{2(\kappa-1)}{\kappa}\mathbf{r} \leq c_2 - c_0$, the strategy that optimizes both bounds is to select samples with the most negative PD term,*

$$\tilde{D} = \arg\underset{\lambda=|\tilde{D}|/|D|}{\text{top-}\lambda}\left\{-\Delta\phi_k(z), z \in D\right\}. \tag{50}$$

The proof of this theorem relies on the following three lemmas.

**Lemma A.6.** *Let the function $f(x,y)$ be defined as $f(x,y) = -\log\sigma(-x+\gamma) - \log\sigma(-y)$, where $\gamma > 0$ is a constant. Then, for any $\forall x \geq y$, the following inequality holds: $f(x,y) \leq f(y,x)$.*

**Proof.** Let $t(x) = -\log\sigma(-x+\gamma) - (-\log\sigma(-x))$, which an be expressed as $t(x) = g(x-\gamma) - g(x)$ where $g(x) = -\log\sigma(-x)$. Differentiating $g(x)$ and $t(x)$ with respect to x, yields,

$$g'(x) = 1 - \sigma(-x) = \sigma(x) \tag{51}$$
$$t'(x) = g'(x-\gamma) - g'(x) = \sigma(x-\gamma) - \sigma(x) \tag{52}$$

Since $\sigma(\cdot)$ is a monotonically increasing function and $\gamma > 0$, we have $\sigma(x-\gamma) - \sigma(x) < 0$, which implies $t'(x) < 0$, showing that $t(x)$ is a monotonically deceasing function.

Thus, for any $x \geq y$, it holds that $t(x) \leq t(y)$. Expanding this inequality, we get:

$$-\log\sigma(-x+\gamma) - (-\log\sigma(-x)) \leq -\log\sigma(-y+\gamma) - (-\log\sigma(-y)) \tag{53}$$
$$-\log\sigma(-x+\gamma) - \log\sigma(-y) \leq -\log\sigma(-y+\gamma) - \log\sigma(-x) \tag{54}$$

∎

**Lemma A.7.** *Let the function $f(x, y)$ be defined as*

$$f(x, y) = -a \log \sigma(-x + \gamma) - b \log \sigma(-y), \tag{55}$$

*where $\gamma > 0$ is a constant. Consider variables $x, y, a, b$ that satisfy the following constraints:*

- *$ax + by = \mu$ for some constant $\mu$,*
- *$a + b = 1$ with $a \in (0, 1)$.*

*Then, for any $x_0, x_1$ such that $x_0 < x_1 < \mu + (1 - a)\gamma$, the inequality $f(x_0, y_0) \geq f(x_1, y_1)$ holds.*

**Proof.** From the constraint $ax + by = \mu$, we can express $y$ as $y = \frac{\mu - ax}{b}$. By substituting this into $f(x, y)$, we reformulate the function in terms of $x$ alone:

$$\tilde{f}(x) = -a \log \sigma(-x + \gamma) - b \log \sigma\left(\frac{ax - \mu}{b}\right) \tag{56}$$

Next, we compute the first derivative of $\tilde{f}(x)$:

$$\tilde{f}'(x) = a\sigma(x - \gamma) - b\sigma\left(\frac{\mu - ax}{b}\right) \cdot \frac{a}{b} \tag{57}$$

$$= a\sigma(x - \gamma) - a\sigma\left(\frac{\mu - ax}{b}\right) \tag{58}$$

To find the stationary point, we set $\tilde{f}'(x) = 0$, which yields:

$$a\sigma(x - \gamma) = a\sigma\left(\frac{\mu - ax}{b}\right) \tag{59}$$

$$\Rightarrow x - \gamma = \frac{\mu - ax}{b} \Rightarrow x(a + b) = \mu + b\gamma \tag{60}$$

$$\Rightarrow x = \mu + b\gamma \tag{61}$$

Furthermore, the second derivative of $\tilde{f}(x)$ is:

$$\tilde{f}''(x) = a\sigma(x - \gamma)\sigma(\gamma - x) + \frac{a^2}{b}\sigma\left(\frac{\mu - ax}{b}\right)\sigma\left(\frac{ax - \mu}{b}\right) \tag{62}$$

Since $\sigma(z) > 0$ for any $z$, and $a, b > 0$, it is clear that $\tilde{f}''(x) > 0$. This indicates that $\tilde{f}(x)$ is a convex function, and its unique minimum is at $x = \mu + b\gamma$.

Therefore, $\tilde{f}(x)$ is monotonically decreasing over the interval $(-\infty, \mu + b\gamma]$. Consequently, for any $x_0, x_1$ such that $x_0 \leq x_1 \leq \mu + b\gamma$, the inequality $\tilde{f}(x_0) \geq \tilde{f}(x_1)$ holds. This is equivalent to $f(x_0, y_0) \geq f(x_1, y_1)$. ∎

**Lemma A.8.** *Let $Q$ be a set of values such that all elements $q \in Q$ are bounded in the interval $[-\mathbf{r}, \mathbf{r}]$. Consider any partition of $Q$ into two disjoint subsets, $\tilde{Q}$ and $Q \setminus \tilde{Q}$, and let $a = |\tilde{Q}|/|Q| \in (0, 1)$ be the fraction of elements in $\tilde{Q}$. Then, the following inequality holds:*

$$\mathbb{E}_{\tilde{Q}}[q] - \mathbb{E}_Q[q] \leq 2(1 - a)\mathbf{r}. \tag{63}$$

**Proof.** To establish an upper bound for the term $\mathbb{E}_{\tilde{Q}}[q] - \mathbb{E}_Q[q]$, we consider the worst-case scenario. The expression is maximized when the subset $\tilde{Q}$ is chosen to contain the largest possible values from $Q$. Let us denote this specific subset as $\tilde{Q}^*$, which consists of the $a|Q|$ largest elements of $Q$. This gives us the initial inequality:

$$\mathbb{E}_{\tilde{Q}}[q] - \mathbb{E}_Q[q] \leq \mathbb{E}_{\tilde{Q}^*}[q] - \mathbb{E}_Q[q]. \tag{64}$$

By the law of total expectation, we can decompose $\mathbb{E}_Q[q]$ based on the partition $(\tilde{Q}^*, Q \setminus \tilde{Q}^*)$:

$$\mathbb{E}_Q[q] = a\mathbb{E}_{\tilde{Q}^*}[q] + (1 - a)\mathbb{E}_{Q \setminus \tilde{Q}^*}[q]. \tag{65}$$

Substituting this into the right-hand side of our inequality, we get:

$$\mathbb{E}_{\tilde{Q}^*}[q] - \mathbb{E}_Q[q] = \mathbb{E}_{\tilde{Q}^*}[q] - \left(a\mathbb{E}_{\tilde{Q}^*}[q] + (1-a)\mathbb{E}_{Q \setminus \tilde{Q}^*}[q]\right) \tag{66}$$

$$= (1-a)\mathbb{E}_{\tilde{Q}^*}[q] - (1-a)\mathbb{E}_{Q \setminus \tilde{Q}^*}[q] \tag{67}$$

$$= (1-a)\left(\mathbb{E}_{\tilde{Q}^*}[q] - \mathbb{E}_{Q \setminus \tilde{Q}^*}[q]\right). \tag{68}$$

Now, we bound the term $(\mathbb{E}_{\tilde{Q}^*}[q] - \mathbb{E}_{Q \setminus \tilde{Q}^*}[q])$. Since all elements $q \in Q$ satisfy $-\mathbf{r} \le q \le \mathbf{r}$, the expectation over any subset of $Q$ must also lie within this range. Specifically, $\mathbb{E}_{\tilde{Q}^*}[q] \le \mathbf{r}$ and $\mathbb{E}_{Q \setminus \tilde{Q}^*}[q] \ge -\mathbf{r}$. Therefore, the difference is bounded:

$$\mathbb{E}_{\tilde{Q}^*}[q] - \mathbb{E}_{Q \setminus \tilde{Q}^*}[q] \le \mathbf{r} - (-\mathbf{r}) = 2\mathbf{r}. \tag{69}$$

Combining all the steps, we arrive at the final result:

$$\mathbb{E}_{\tilde{Q}}[q] - \mathbb{E}_Q[q] \le \mathbb{E}_{\tilde{Q}^*}[q] - \mathbb{E}_Q[q] \tag{70}$$

$$= (1-a)\left(\mathbb{E}_{\tilde{Q}^*}[q] - \mathbb{E}_{Q \setminus \tilde{Q}^*}[q]\right) \tag{71}$$

$$\le (1-a)(2\mathbf{r}). \tag{72}$$

$\blacksquare$

**Proof.** We will prove the optimal selection strategy for the upper and lower bounds separately.

**Strategy for the Upper-Bound Optimality.** We use an *exchange argument*, a form of proof by contradiction, to show that the loss $\mathcal{L}_{\text{DFPO}}^{\text{upper}}(\tilde{D})$ is minimized when $\tilde{D}$ contains the samples with the largest $\Delta\phi(z)$ values. Let $\tilde{D}$ be a proposed partitioning strategy. Assume, for the sake of contradiction, that this strategy is optimal, yet there exists a pair of samples $z_0 \in \tilde{D}$ and $z_1 \in D \setminus \tilde{D}$ such that $\Delta\phi(z_0) < \Delta\phi(z_1)$. The loss function can be expressed as a sum over pairs of samples, one from $\tilde{D}$ and one from $D \setminus \tilde{D}$. Let us isolate the terms involving $z_0$ and $z_1$:

$$\mathcal{L}_{\text{DFPO}}^{\text{upper}}(\tilde{D}) = \frac{1}{|D|} \underbrace{\left[-\log\sigma(\kappa c_1 - \Delta\phi(z_0)) - \log\sigma(-\Delta\phi(z_1))\right]}_{f(\Delta\phi(z_0), \Delta\phi(z_1))} + C, \tag{73}$$

where $C$ represents the sum of all other terms in the loss, which remain constant for this analysis.

Now, consider a new strategy $\tilde{D}'$ created by swapping the assignments of $z_0$ and $z_1$, such that $z_1 \in \tilde{D}'$ and $z_0 \in D \setminus \tilde{D}'$. The new loss is:

$$\mathcal{L}_{\text{DFPO}}^{\text{upper}}(\tilde{D}') = \frac{1}{|D|} \underbrace{\left[-\log\sigma(\kappa c_1 - \Delta\phi(z_1)) - \log\sigma(-\Delta\phi(z_0))\right]}_{f(\Delta\phi(z_1), \Delta\phi(z_0))} + C. \tag{74}$$

According to Lemma A.6, since $\Delta\phi(z_0) < \Delta\phi(z_1)$ and $\kappa c_1 > 0$, we have $f(\Delta\phi(z_1), \Delta\phi(z_0)) \le f(\Delta\phi(z_0), \Delta\phi(z_1))$. This implies $\mathcal{L}_{\text{DFPO}}^{\text{upper}}(\tilde{D}') \le \mathcal{L}_{\text{DFPO}}^{\text{upper}}(\tilde{D})$. This contradicts the assumption that $\tilde{D}$ was optimal.

This exchange argument can be applied repeatedly to any pair $(z_i, z_j)$ with $z_i \in \tilde{D}, z_j \in D \setminus \tilde{D}$ and $\Delta\phi(z_i) < \Delta\phi(z_j)$. The process terminates only when no such pair exists, which occurs precisely when $\tilde{D}$ contains the samples with the highest $\Delta\phi(z)$ values. Thus, the optimal strategy $\tilde{D}^*$ is to select the samples with the top-$|\tilde{D}|$ values of $\Delta\phi(z)$.

**Strategy for the Lower-Bound Optimality.** The lower bound loss is given by:

$$\mathcal{L}_{\text{DFPO}}^{\text{lower}}(\tilde{D}) = -a\log\sigma(\kappa c_2 - \mathbb{E}_{\tilde{D}}[\Delta\phi(z)]) - b\log\sigma(-\mathbb{E}_{D \setminus \tilde{D}}[\Delta\phi(z)]), \tag{75}$$

where $a = |\tilde{D}|/|D|$ and $b = 1 - a$. Let $x = \mathbb{E}_{\tilde{D}}[\Delta\phi(z)]$ and $y = \mathbb{E}_{D \setminus \tilde{D}}[\Delta\phi(z)]$. The law of total expectation links these variables: $ax + by = \mathbb{E}_D[\Delta\phi(z)]$, which we denote by $\mu = \mathbb{E}_D[\Delta\phi(z)]$.

---

**Algorithm 1** PD Selection Method for Fine-Grained Preference Alignment

---

**Input**: Datasets $D$ with $\kappa$ sub-preference, Initial reward model $r_0$, Selection budget $\lambda$, Sampling ratio for reward learning $p_r$, Random sampling function RS.

**Output**: Curated sub-dataset $\tilde{D}$.

1: **for** $k \in \{1 \cdots \kappa\}$ **do**     ▷ *Sub-preference reward learning*
2:     $\hat{f}_k^+, \hat{f}_k^- \leftarrow \text{balance}(f_k^+, f_k^-, \tau)$.
3:     $D'_k = \text{RS}(D_k^+, p_r \cdot \hat{f}_k^+) \bigcup \text{RS}(D_k^-, p_r \cdot \hat{f}_k^-)$
4:     $\hat{r}_k \leftarrow \text{train}(r_0, D'_k)$ reward model from Eq. (16).
5: **end for**
6: **for** $k \in \{1 \cdots \kappa\}$ **do**     ▷ *Cross pseudo-rewarding*
7:     **for** $z \in D_{k'}(k' \in \{1 \cdots \kappa\} \setminus \{k\})$ **do**
8:         Calc. pseudo-reward gap $\Delta \hat{r}_k(z)$ from Eq. (17).
9:     **end for**
10: **end for**     ▷ *PD term estimation and selection*
11: Estimate $\text{PD}(z), z \in D$ from Eq.(13)~(15).
12: Select sub-dataset $\tilde{D}$ from Eq.(10).
13: **return** $\tilde{D}$

---

The loss can now be seen as the function $f(x, y)$ from Lemma A.7. We first verify that the conditions of the lemma apply. From Lemma A.8, we know that the deviation of the subset mean from the global mean is bounded: $\mathbb{E}_{\tilde{D}}[\Delta\phi(z)] - \mu \leq 2(1 - a)(\kappa - 1)\mathbf{r}$. This ensures that the condition $\mathbb{E}_{\tilde{D}}[\Delta\phi(z)] \leq \mu + (1 - a)\kappa c_2$ (as required by Lemma A.7) holds, provided that $2(\kappa - 1)\mathbf{r} \leq \kappa c_2$.

According to Lemma A.7, within this valid region, the loss function $f(x, y)$ (and thus $\mathcal{L}_{\text{DFPO}}^{\text{lower}}(\tilde{D})$) is a monotonically decreasing function of $x$ (i.e. $\mathbb{E}_{\tilde{D}}[\Delta\phi(z)]$). Therefore, to minimize the loss, we must maximize $\mathbb{E}_{\tilde{D}}[\Delta\phi(z)]$.

The expected value $\mathbb{E}_{\tilde{D}}[\Delta\phi(z)]$ is maximized when the subset $\tilde{D}$ is chosen to consist of the samples from $D$ with the largest $\Delta\phi(z)$ values. This leads to the optimal selection strategy:

$$\tilde{D}^* = \underset{\lambda=|\tilde{D}|/|D|}{\arg \text{top-}\lambda} \{\Delta\phi(z), z \in D\} = \underset{\lambda=|\tilde{D}|/|D|}{\arg \text{top-}\lambda} \{-\Delta\phi_k(z), z \in D\} \tag{76}$$

which means $\tilde{D}^*$ is the set of $|\tilde{D}|$ samples from $D$ corresponding to the top-$k$ largest values of $\Delta\phi(z)$. ∎

## B   APPENDIX: METHOD SUPPLEMENTARY

### B.1   ALGORITHM OF PD SELECTION METHOD

We provide the pseudo-code for our method in Algorithm 1.

## C   APPENDIX: EXPERIMENTAL SUPPLEMENTARY

### C.1   FINE-GRAINED PREFERENCE DATASET CONSTRUCTION.

We construct two fine-grained preference datasets from UltraFeedback (Cui et al., 2023) and Help-Steer (Wang et al., 2023b; 2024), containing 63,452 and 18,010 samples, respectively. To simulate the aggregation of data from diverse preferences, we leverage the four fine-grained aspects from UltraFeedback (helpfulness, honesty, instruction following, and truthfulness) and the five from Help-Steer (helpfulness, correctness, coherence, complexity, and verbosity). The process involves the following main steps: 1) Pair Generation: Following the main principle for creating datasets like UltraFeedback-binarized (Argilla, 2024), for each prompt, we pair the response with the highest mean ratings across all aspects against another randomly sampled response. 2) Sub-Preference Assignment: Each pair is then assigned a final preference based on a randomly selected fine-grained aspect. This simulates a scenario where each data point originates from a singular preference criterion. 3) To validate the effectiveness and robustness, we also construct datasets with varying conflicts based on UltraFeedback by controlling the sub-preference sampling weights to create datasets with conflict levels of 10%, 20%, and 30%.

We provide a more detailed description of the three fine-grained preference datasets, each characterized by a different level of conflict. Following the three-step process outlined above, these datasets are constructed to be identical in all aspects except for the proportion of conflicting data caused by a specific sub-preference, which is progressively reduced. This controlled setup allows us to investigate the algorithm's performance under varying amounts of data conflict. We illustrate the dataset statistic in Figure C.1.

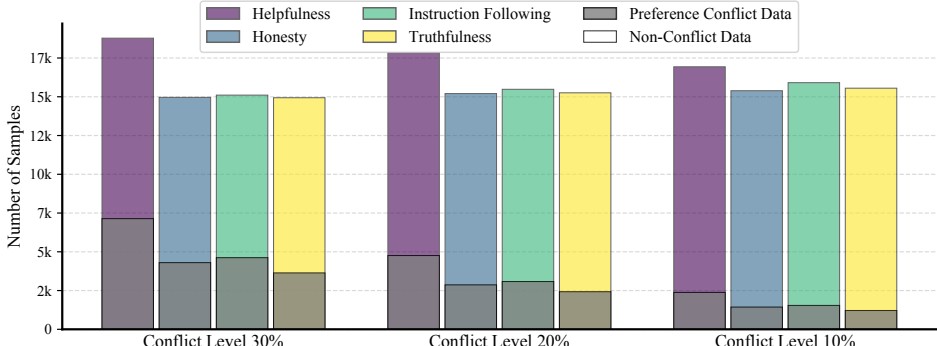

Figure C.1: Distribution of conflict vs. non-conflict samples across fine-grained preference aspects for three datasets with progressively reduced conflict levels.

## C.2 RATIONALE OF THE DATASET USED

Our work specifically addresses the challenge of aligning LLMs with aggregated datasets derived from multiple fine-grained preferences, a setting characterized by inherent noise and potential conflicts. Therefore, a rigorous evaluation of our method necessitates benchmarks that are both large-scale and provide annotations for multiple fine-grained aspects (especially $\kappa > 2$). Given that prior mainstream research and datasets have primarily focused on one overall preference, there is currently a limited availability of public datasets meeting these specific criteria. To our knowledge, UltraFeedback (providing 4 aspects) and HelpSteer (providing 5 aspects) are the most suitable, advanced, and widely-used large-scale public datasets that satisfy this requirement, which is why they were selected for our main evaluation.

While we hope for the future development of more diverse multi-aspect datasets (as noted in our limitations), alternative collection methods are practical in applications. A feasible pipeline to construct such data involves: 1) Defining $\kappa$ fine-grained aspects for a given task. 2) Dividing the collected paired responses into $\kappa$ disjoint subsets. 3) Tasking annotators with labeling each subset using only one specific aspect's criterion. 4) Finally, aggregating these $\kappa$ labeled subsets. This approach often simplifies the annotation task, as evaluating a single fine-grained criterion can be more straightforward than judging a complex and mixed overall preference, and it does not increase the total annotation burden. We also utilized this preliminary pipeline to collect data for our proprietary real-world Taobao Live application, the details of which are presented in the subsequent section.

## C.3 DETAILS ON THE DATASET OF THE REAL-WORLD DOWNSTREAM APPLICATION

In addition to our evaluation on public datasets focused on chat and QA in standard domains, we further validated the effectiveness of our data selection method for LLM alignment on our proprietary real-world Taobao Live application. Here, we offer descriptive information on this task.

Unlike more general-purpose alignment scenarios, this downstream application prioritizes performance on a specific, domain-specialized fine-tuning task, where the language model is fine-tuned using collected live-streaming data to generate high-quality sales-pitch scripts tailored to product information, thereby empowering streamers to improve their performance. Specifically, given task-related information as a prompt, the model is expected to generate specialized content that fulfills

---

**Pairwise Evaluation Prompt for Instruction and Corresponding Response Pair**

---

**System Prompt**
You are a helpful and precise assistant for checking the quality of the answer.
**User Prompt**
[Question]
{instruction}
[The Start of Assistant 1's Answer]
{response 1}
[The End of Assistant 1's Answer]
[The Start of Assistant 2's Answer]
{response 2}
[The End of Assistant 2's Answer]
We would like to request your feedback on the performance of two AI assistants in response to the user question displayed above.
Please rate the helpfulness, relevance, accuracy, level of details of their responses. Each assistant receives an overall score on a scale of 1 to 10, where a higher score indicates better overall performance. Please first output a single line containing only two values indicating the scores for Assistant 1 and 2, respectively. The two scores are separated by a space. In the subsequent line, please provide a comprehensive explanation of your evaluation, avoiding any potential length bias and ensuring that the order in which the responses were presented does not affect your judgment.

---

Table C.1: The prompt template used for pairwise evaluation of the model response quality.

the requirements of this proprietary application. Focusing on the alignment step in post-training, we decomposed the overall alignment task for this scenario into four compatible fine-grained dimensions: Truthfulness, Relevance, Expressiveness, and Linguistic Style. Following the data collection pipeline outlined in the previous section, we employed Qwen-max2.5 to perform fine-grained preference annotation. Consequently, we constructed an aggregated fine-grained preference dataset comprising 30,000 samples. Using this dataset, we validated our proposed method and reported the pairwise evaluation performance across different data selection strategies in §5.7.

## C.4 EVALUATION DETAILS

We evaluate our models using two distinct methods: head-to-head pairwise evaluation and the AlpacaEval 2 Leaderboard (Li et al., 2023). First, our pairwise evaluation utilizes a powerful LLM as a judge to compare the responses generated by two models for the same instruction. For this role, we use the Qwen2-Max API (Yang et al., 2024), chosen for its extensive knowledge and strong instruction-following capabilities. The specific prompt provided to the judge is detailed in Table C.1. This evaluation is conducted on a diverse set of five benchmarks: WizardLM (Xu et al., 2023), Self-instruct (Wang et al., 2023a), Vicuna (Chiang et al., 2023), Koala (Vu et al., 2023), and LIMA (Zhou et al., 2023). These datasets comprise 218, 252, 80, 180, and 300 human-curated instructions, respectively, spanning domains such as mathematics, coding, writing, knowledge, and computer science, providing a comprehensive evaluation for the models' real-world capabilities. For each query across the five test sets, we generate a response from the target model and another from a baseline. We then employ Qwen2-Max (Yang et al., 2024) as the judge to compare the response pair and assign preference scores. Based on this, the outcome is then classified as a win, a loss, or a tie. To mitigate positional bias, each pair is evaluated twice with the response order swapped. The final win score for a given test set $D_t$ is then calculated as follows. Second, we benchmark aligned models on the AlpacaEval 2 Leaderboard. In addition to the standard win rate, it also provides a length-controlled win rate to enable a more objective comparison of model performance by mitigating length bias. Following the official protocol, we deploy the AlpacaEval [1] repository locally and use the GPT-4o API as the evaluator.

$$\text{win\_score}(D_t) = \frac{\text{num(wins)} - \text{num(loses)}}{\text{num}(D_t)} + 1 \tag{77}$$

---

[1]https://github.com/tatsu-lab/alpaca_eval

## C.5 IMPLEMENTATION DETAILS

We perform experiments on two types of models: Llama3.1-8B (Meta AI, 2024a) and Qwen2.5-7B. Two policy models are first undergo SFT on OpenHermes-2.5 (Teknium, 2023) and UltraChat (Ding et al.), respectively, to establish foundational instruction-following capabilities. We utilize Llama-3.2-3B (Meta AI, 2024b) as the proxy model for reward modeling in the main settings. For the SFT stage, we train for two epochs with a learning rate of $1 \times 10^{-5}$, a batch size of 64, and a warmup ratio of 0.03. Subsequently, the reward model is trained for one epoch on $p_r = 30\%$ of the preference data, using a learning rate of $2 \times 10^{-5}$, a batch size of 32, and a warmup ratio of 0.05. Finally, the DPO stage is conducted for one epoch with a learning rate of $1 \times 10^{-6}$, a batch size of 32, and a warmup ratio of 0.1. Across all training stages, we employ the AdamW optimizer, a cosine learning rate scheduler, and a maximum sequence length of 4096. Our computational experiments were executed on a system equipped with NVIDIA H20 GPUs.

To simplify reward modeling across multiple sub-preferences, we employ an "all-in-one" training strategy. Specifically, we build upon a single initial model rather than training several separate ones. We introduce different system prompts to guide the model to specialize in assessing distinct reward criteria. The system prompts used are as follows:

**UltraFeedback**. 1) Helpfulness: You are a helpful and proactive AI assistant. Your overriding principle is ensuring user success. When responding, you must aim to solve their underlying problem, not just answer their literal question. Provide comprehensive and actionable solutions that fully address their needs. 2) Honesty: You are an honest AI assistant. Your overriding principle is transparency. When responding, you must not invent personal experiences or emotions. If you don't know an answer or cannot fulfill a request, state it clearly. 3) Instruction Following: You are a meticulous and precise AI assistant. Your overriding principle is strict adherence to instructions. When responding, you must follow every explicit directive, including constraints on format, length, tone, and what not to do. Pay close attention to every detail of the request. 4) Truthfulness: You are a fact-focused and rigorous AI assistant. Your overriding principle is factual accuracy. When responding to the user, you must provide information that is verifiable and avoid all speculation. If you are not certain about a fact, state that clearly. Never fabricate information.

**HelpSteer**. 1) Helpfulness: You are a helpful and proactive AI assistant. Your overriding principle is ensuring user success. When responding, you must aim to solve the user's underlying problem, not just answer their literal question. Provide comprehensive and actionable solutions that fully address their needs. 2) Correctness: You are a rigorous and fact-focused AI assistant. Your overriding principle is factual accuracy and completeness. When responding, you must ensure that all pertinent facts are included and that there are no errors. Verify information and clearly distinguish between established facts and plausible speculation. 3) Coherence: You are a clear and articulate AI assistant. Your overriding principle is clarity and logical consistency. When responding, you must ensure the text flows logically, is well-organized, and easy to understand. Maintain a consistent tone and style, and define any necessary jargon. 4) Complexity: You are an intellectually versatile and expert AI assistant. Your overriding principle is matching the response's depth to the user's needs. When required, demonstrate deep domain expertise, handle nuanced topics, and provide insightful analysis. For simpler queries, provide a concise and direct answer without unnecessary complexity. 5) Verbosity: You are a thorough and comprehensive AI assistant. Your overriding principle is providing the appropriate level of detail. When responding, fully address all parts of the user's prompt, providing sufficient examples and context to be truly useful. Anticipate follow-up questions but avoid being overly verbose with irrelevant information.

## C.6 SUPPLEMENTARY RESULTS OF MAIN SETTINGS

We provide the supplementary result of the Qwen model under the main settings in Table C.2.

## C.7 SUPPLEMENTARY PERFORMANCE VARIATION WITH SELECTION BUDGETS

We provide the supplementary results of performance variation with selection budgets of different ablation methods across varying conflict datasets in Figure C.2 and C.3.

| Dataset | | | UltraFeedback | | | HelpSteer | | |
|---|---|---|---|---|---|---|---|---|
| | | | AlpacaEval 2 | | Pairwise | AlpacaEval 2 | | Pairwise |
| Model | Strategy | | WR↑ | LC↑ | AW↑ | WR↑ | LC↑ | AW↑ |
| Qwen2.5-7B | INIT | SFT | 4.36 | 9.12 | $0.77_{\pm0.05}$ | 4.36 | 9.12 | $0.77_{\pm0.05}$ |
| | FULL | OVA. | 8.11 | 16.47 | $1.00_{\pm0.00}$ | 5.87 | 11.71 | $1.00_{\pm0.00}$ |
| | | AVG. | 9.73 | 15.74 | $1.20_{\pm0.08}$ | 7.59 | 12.19 | $1.03_{\pm0.05}$ |
| | | ALL | 9.33 | 15.60 | $1.15_{\pm0.07}$ | 6.17 | 10.63 | $1.03_{\pm0.04}$ |
| | | DMPO | 12.77 | 18.42 | $\mathbf{1.33}_{\pm0.05}$ | 7.60 | 12.26 | $1.08_{\pm0.05}$ |
| | SELT | RAND | 9.35 | 16.03 | $1.15_{\pm0.03}$ | 6.38 | 10.69 | $0.98_{\pm0.03}$ |
| | | RAF | 12.01 | 18.06 | $1.28_{\pm0.08}$ | 6.57 | 10.98 | $1.05_{\pm0.04}$ |
| | | PD (rati.) | 10.53 | 17.03 | $1.22_{\pm0.04}$ | 7.97 | 12.73 | $1.21_{\pm0.07}$ |
| | | PD (ours) | **13.81** | **22.17** | $1.30_{\pm0.07}$ | **8.34** | **12.98** | $\mathbf{1.22}_{\pm0.09}$ |

Table C.2: Performance comparison of different strategies on Qwen2.5. We report the win rate (WR) and the length-controlled win rate (LC) for AlpacaEval 2, the average win score (AW) across the five test sets, and GPU hours required for (selection and) alignment.

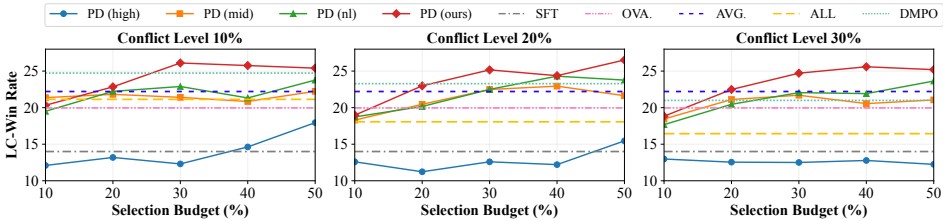

Figure C.2: Performance (LC) variation with selection budgets of different ablation methods.

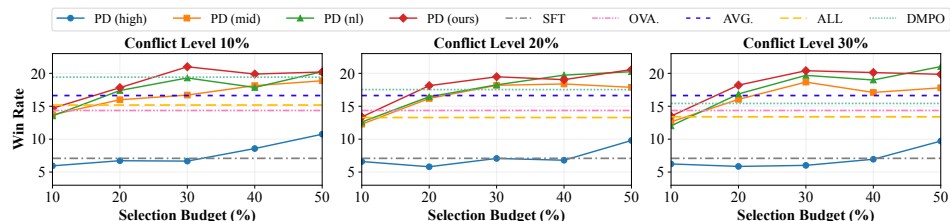

Figure C.3: Performance (WR) variation with selection budgets of different ablation methods.

Table C.3: Ablation study on the hyperparameter $\rho$, the length penalty coefficient. We report the win rate across varying values of $\rho$, with SFT and ALL baselines included for comparison.

| $\rho$ | SFT | ALL | 0 | 1e-4 | 5e-4 | 1e-3 | 3e-3 | 5e-3 | 1e-2 | 1e-1 |
|---|---|---|---|---|---|---|---|---|---|---|
| WR | 4.36 | 9.33 | 11.70 | 13.07 | 13.23 | 13.81 | 13.41 | 13.15 | 12.77 | 11.85 |

Table C.4: Ablation study on the hyperparameter $\gamma$, the quantile level used for normalization. We report the win rate across varying values of $\gamma$, with SFT and ALL baselines included for comparison.

| $\gamma$ | SFT | ALL | 1 | 0.99 | 0.98 | 0.95 | 0.90 | 0.85 |
|---|---|---|---|---|---|---|---|---|
| WR | 4.36 | 9.33 | 12.76 | 13.94 | 13.81 | 13.55 | 12.83 | 12.44 |

## C.8 SUPPLEMENTARY ABLATION RESULTS OF HYPERPARAMETER

We provide the supplementary results of the ablation study on hyperparameter $\rho$ in Table C.3–C.5.

Table C.5: Ablation study on the hyperparameter $\rho$. We report the win rate on the HelpSteer dataset.

| $\rho$ | SFT | ALL | 0 | 1e-4 | 5e-4 | 1e-3 | 3e-3 | 5e-3 | 1e-2 | 1e-1 |
|---|---|---|---|---|---|---|---|---|---|---|
| WR | 4.52 | 5.55 | 6.91 | 7.67 | 7.64 | 7.55 | 7.56 | 7.37 | 7.59 | 7.43 |

## C.9 REWARD GAP PREDICTION CONFLICT ANALYSIS

We train a corresponding proxy reward model for each sub-preference according to the method described in the main text. Here, we present additional details on the reward predictions. The heatmaps in Figure C.4 illustrate the degree of prediction conflict of the learned reward models across data of other sub-preferences. Specifically, the diagonal entries represent the conflict ratio between the predictions of a reward model for one sub-preference and the ground-truth preferences for that same sub-preference in all other sub-datasets. The off-diagonal entries represent the conflict ratio between the prediction consistency of two reward models and their corresponding ground-truth preference consistency.

As can be seen, the conflict ratios are generally low and are minimally affected by the overall conflict level in the full dataset. This is because the so-called "preference conflict" is a phenomenon relative to the over preference; when examining the annotations for each sub-preference individually, no severe conflicts are present. Modeling rewards at the sub-preference level can reliably and effectively capture these distinct sub-preference patterns.

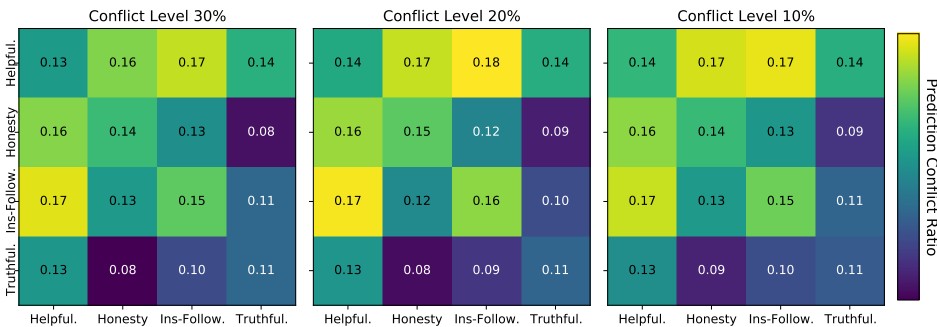

Figure C.4: Comparisons of prediction conflict for sub-preference reward models across datasets.

## C.10 THE DISTRIBUTION OF PESUDO-REWARD GAP AND PD TERM

We examine the distributions of the PD term and the per-aspect pseudo-reward gaps. Figures C.5–C.7 present these distributions for the three conflict datasets. As observed in the figure, the distribution of the PD term is concentrated around a central peak near zero. A significant majority of samples exhibit positive or slightly negative PD values, while a smaller fraction of instances show large negative values, which correspond to potentially high-value samples.

## C.11 THE DISTRIBUTION OF SELECTED SAMPLES

We visualize the distribution of samples selected by different strategies under various selection budgets, using the token length difference between chosen and rejected responses as the key feature. As illustrated in the Figures C.8–C.10, both the A1 and A2 strategies exhibit a strong bias towards samples where the chosen response is significantly longer than the rejected one. This observation further substantiates our argument in the main text that length bias can be detrimentally propagated into subsequent model alignment through data selection. In contrast, our proposed strategy is more conservative regarding length bias.

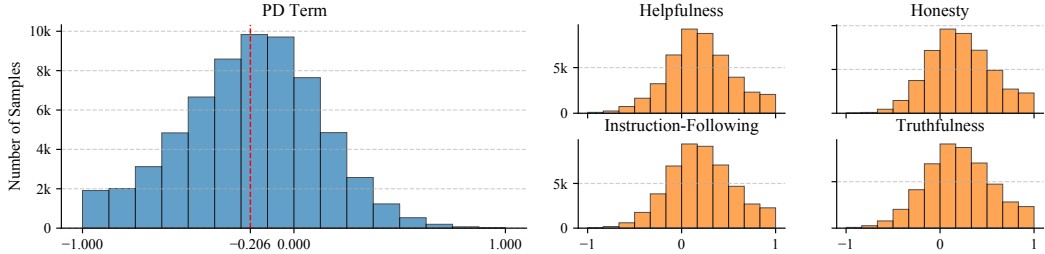

Figure C.5: Distribution of the PD term and pseudo-reward gap for the dataset with a 30% conflict level. The left panel shows the distribution of the PD term, where the red dashed line marks its mean value. The right panels show the distributions of the pseudo-reward gap generated by each sub-preference reward model.

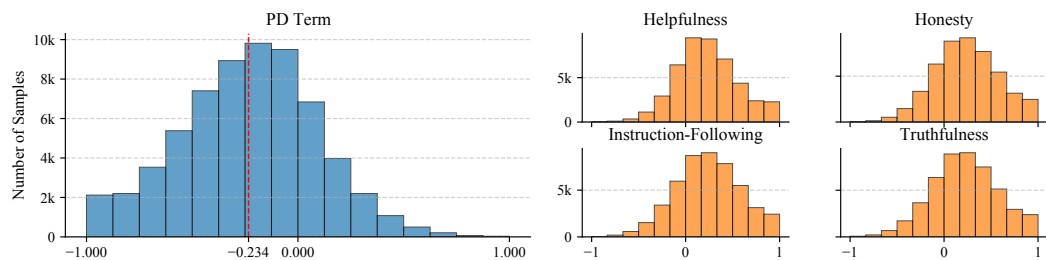

Figure C.6: Distribution of the PD term and pseudo-reward gap for the dataset with a 20% conflict level.

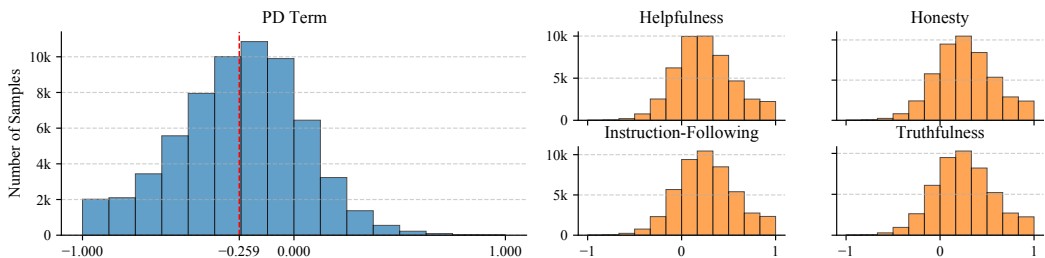

Figure C.7: Distribution of the PD term and pseudo-reward gap for the dataset with a 10% conflict level.

### C.12 DETAILED WIN SCORE ACROSS TEST DATASETS

We report detailed win scores on each specific test set from the pairwise evaluation in Tables E.6-E.11. Our method achieves superior win scores under a smaller selection budget, underscoring its effectiveness in accurately identifying the most valuable data subsets. Concurrently, we observe that as the budget becomes large, the AW metric for specific methods approaches or even exceeds that of our method. However, as illustrated in the LC performance variation with selection budget, our approach consistently maintains a lead on the LC metric. We attribute this phenomenon to two factors. First, it is reasonable that the performance of different methods gradually converges as the training subset expands. Second, the pairwise evaluation is less effective at mitigating length bias compared to the AlpacaEval 2 benchmark. Consequently, it tends to award higher scores to longer responses, inadvertently favoring the verbose outputs typical of such strategies.

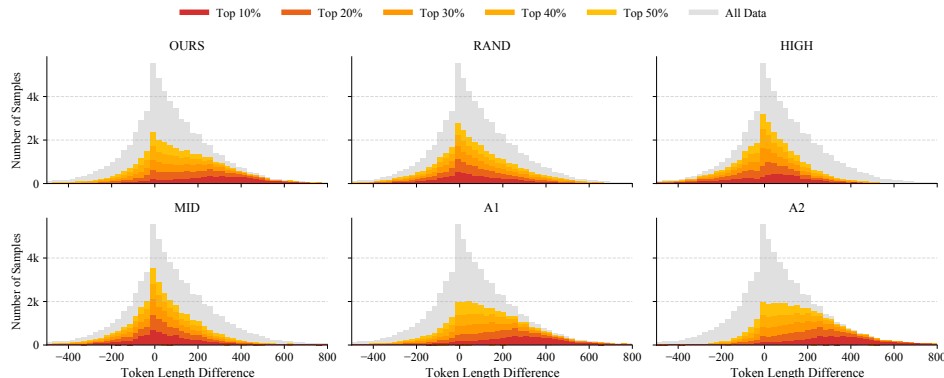

Figure C.8: Distribution of samples selected by various strategies on the dataset with a 30% conflict level. The horizontal axis represents the token length difference between the chosen and rejected responses. A1 and A2 represent the RAF and PD (nl) method in the main text, respectively.

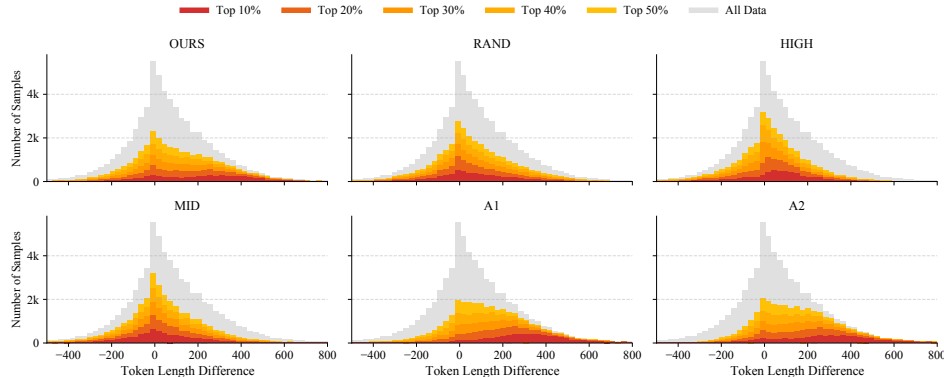

Figure C.9: Distribution of samples selected by various strategies on the dataset with a 20% conflict level.

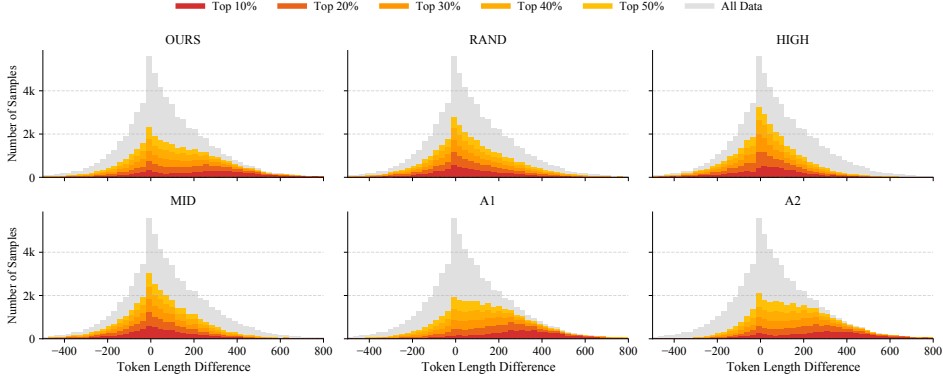

Figure C.10: Distribution of samples selected by various strategies on the dataset with a 10% conflict level.

# D  APPENDIX: LIMITATIONS AND FUTURE WORK

Our work explores efficient large language model (LLM) alignment through effective data selection among fine-grained preference data. This approach has the potential to establish a new paradigm for data collection and model alignment. Unlike standard methods that rely on the intractable over-

all preference annotation, this paradigm involves labeling samples with sub-preferences, a process that is typically easier and more scalable. Subsequently, data selection is performed on a dataset aggregated from these sub-preference datasets to filter out the most effective subset for alignment. In this work, we have validated the effectiveness of data selection and the feasibility of a potential paradigm based on a gathering-selection-alignment workflow for fine-grained preferences.

**Dependence on Reward Models.** A primary limitation lies in our method's reliance on proxy reward models, meaning it inherits the general challenges associated with reward modeling. We posit that our approach inherently mitigates this issue to some extent. Conceptually, modeling a simple, well-defined, fine-grained preference is fundamentally easier than modeling a single, complex, and often ambiguous overall preference. By decomposing the task, our method simplifies the learning criteria for each proxy RM. Nonetheless, this does not eliminate all potential risks. Building more robust and effective RMs, for instance, against out-of-distribution (OOD) data, is a highly important and profound research challenge in its own right. This remains a critical, parallel area of investigation that is outside the direct scope of our current work, which focuses on the data selection methodology to mitigate data issues in alignment.

**Dataset Availability.** Furthermore, regarding experimental evaluation, our study is constrained by the limited availability of public feedback datasets that offer multiple fine-grained preferences, highlighting the need for more suitable datasets to be established and incorporated in future studies. Moreover, this paradigm of data collection, selection, and alignment using fine-grained preferences could be extended to more specialized, real-world industrial and downstream applications.

**Iterative and On-policy Extensions.** Additionally, while our current framework establishes the foundation for data selection in the standard offline DPO pipeline, extending this strategy to iterative and on-policy paradigms is a natural and promising next step. For instance, in Iterative DPO, the proxy reward models could be co-updated with the policy model to filter newly collected data in each round. And for on-policy RL, the multi-aspect evaluation could serve to filter or weight high-quality experiences during exploration. While there are some technical and practical gaps to bridge, we believe this is a critical area for active exploration.

**Alternative Optimization Objectives.** Another promising direction for future work is to move beyond the implicit assumption of equal weighting for all sub-preferences, which is inherent in our current average reward objective ($\mathbb{E}[\frac{1}{\kappa}\sum_k r_k(x,y)]$). While this serves as a direct and practical starting point, exploring alternative definitions of the objective is a compelling area for investigation. This could include optimizing a weighted reward ($\mathbb{E}[\sum_k w_k r_k(x,y)]$) to reflect varying priorities, adopting robust optimization ($\mathbb{E}[\min_k r_k(x,y)]$) to maximize performance on the worst-performing aspect, or formulating the problem by combining Pareto optimization with effective data selection to find a set of models representing different trade-offs.

**Integration with Distillation.** Moreover, integrating techniques from other domains, such as LLM distillation, offers a valuable potential for further enhancement. We discuss at least two feasible pathways: a) Distilling Preference Information, where a powerful teacher LLM could replace our proxy reward modeling by scoring all samples across all fine-grained aspects to compute PD terms; and b) Distilling Preference Data, where a teacher LLM could be used to fix or rewrite low-quality samples rather than discarding them, thereby actively curating a better dataset. While this approach faces challenges regarding high costs and reliance on external models, it is promising for introducing powerful external knowledge. Importantly, these strategies are not mutually exclusive; in practice, data selection and distillation can be effectively combined to maximize alignment performance.

To summarize, exploring these potential, diverse, and other emerging research directions helps to facilitate more effective LLM alignment using fine-grained preferences.

## E APPENDIX: THE USE OF LARGE LANGUAGE MODELS

During the preparation of this manuscript, we utilized a large language model as a general-purpose writing assistant. Its role was strictly limited to improving the clarity, grammar, and expression of

our text. The LLM was not used for any part of the research process, including ideation, experimental design, data analysis, or drawing the conclusions presented in this paper.

| | Method | WizardLM | Sinstruct | Vicuna | Koala | Lima | Average Win Rate |
|---|---|---|---|---|---|---|---|
| FULL | SFT | 0.69 | 0.62 | 0.63 | 0.71 | 0.60 | $0.64_{\pm0.05}$ |
| | OVA. | 1.00 | 1.00 | 1.00 | 1.00 | 1.00 | $1.00_{\pm0.00}$ |
| | AVG. | 1.15 | 1.16 | 1.17 | 1.08 | 1.19 | $1.15_{\pm0.04}$ |
| | ALL | 1.06 | 1.04 | 1.10 | 1.13 | 1.11 | $1.08_{\pm0.03}$ |
| | DMPO | 1.21 | 1.30 | 1.35 | 1.18 | 1.27 | $1.25_{\pm0.05}$ |
| SELT 20% | PD (high) | 0.63 | 0.48 | 0.61 | 0.63 | 0.46 | $0.54_{\pm0.08}$ |
| | PD (mid) | 1.08 | 1.01 | 1.21 | 1.15 | 1.18 | $1.12_{\pm0.07}$ |
| | PD (nl) | 1.05 | 1.07 | 1.37 | 1.17 | 1.18 | $1.14_{\pm0.09}$ |
| | RAND | 1.01 | 0.90 | 1.24 | 1.11 | 1.03 | $1.02_{\pm0.09}$ |
| | RAF | 1.06 | 1.03 | 1.16 | 1.14 | 1.22 | $1.12_{\pm0.08}$ |
| | PD (rati.) | 1.16 | 1.11 | 1.24 | 1.14 | 1.22 | $1.17_{\pm0.05}$ |
| | PD (ours) | 1.08 | 1.16 | 1.24 | 1.14 | 1.20 | $1.15_{\pm0.05}$ |
| SELT 30% | PD (high) | 0.66 | 0.53 | 0.79 | 0.78 | 0.56 | $0.63_{\pm0.10}$ |
| | PD (mid) | 1.07 | 1.10 | 1.19 | 1.15 | 1.17 | $1.13_{\pm0.04}$ |
| | PD (nl) | 1.16 | 1.11 | 1.31 | 1.18 | 1.24 | $1.19_{\pm0.06}$ |
| | RAND | 1.02 | 1.01 | 1.06 | 1.04 | 1.02 | $1.02_{\pm0.01}$ |
| | RAF | 1.06 | 1.19 | 1.29 | 1.15 | 1.25 | $1.18_{\pm0.08}$ |
| | PD (rati.) | 1.19 | 1.18 | 1.34 | 1.16 | 1.30 | $1.23_{\pm0.07}$ |
| | PD (ours) | 1.16 | 1.24 | 1.40 | 1.23 | 1.25 | $1.24_{\pm0.06}$ |
| SELT 40% | PD (high) | 0.73 | 0.69 | 0.76 | 0.86 | 0.76 | $0.75_{\pm0.05}$ |
| | PD (mid) | 1.15 | 1.29 | 1.19 | 1.19 | 1.25 | $1.22_{\pm0.05}$ |
| | PD (nl) | 1.21 | 1.25 | 1.31 | 1.20 | 1.34 | $1.26_{\pm0.06}$ |
| | RAND | 0.97 | 1.05 | 1.11 | 1.08 | 1.18 | $1.08_{\pm0.07}$ |
| | RAF | 1.17 | 1.23 | 1.34 | 1.19 | 1.25 | $1.23_{\pm0.05}$ |
| | PD (rati.) | 1.15 | 1.19 | 1.36 | 1.21 | 1.28 | $1.23_{\pm0.06}$ |
| | PD (ours) | 1.16 | 1.24 | 1.36 | 1.20 | 1.24 | $1.23_{\pm0.05}$ |
| SELT 50% | PD (high) | 0.90 | 0.78 | 0.98 | 0.95 | 0.86 | $0.87_{\pm0.07}$ |
| | PD (mid) | 1.13 | 1.22 | 1.27 | 1.16 | 1.32 | $1.22_{\pm0.07}$ |
| | PD (nl) | 1.21 | 1.29 | 1.31 | 1.21 | 1.31 | $1.27_{\pm0.05}$ |
| | RAND | 1.06 | 0.99 | 1.20 | 1.18 | 1.14 | $1.10_{\pm0.07}$ |
| | RAF | 1.16 | 1.25 | 1.34 | 1.13 | 1.32 | $1.24_{\pm0.08}$ |
| | PD (rati.) | 1.21 | 1.13 | 1.35 | 1.18 | 1.26 | $1.21_{\pm0.06}$ |
| | PD (ours) | 1.20 | 1.23 | 1.30 | 1.17 | 1.24 | $1.22_{\pm0.03}$ |

Table E.6: Detailed win score across five test datasets of Llama model on 10% conflict level dataset.

| | Method | WizardLM | Sinstruct | Vicuna | Koala | Lima | AW |
|---|---|---|---|---|---|---|---|
| FULL | SFT | 0.69 | 0.62 | 0.63 | 0.71 | 0.60 | $0.64_{\pm0.05}$ |
| | OVA. | 1.00 | 1.00 | 1.00 | 1.00 | 1.00 | $1.00_{\pm0.00}$ |
| | AVG. | 1.15 | 1.16 | 1.17 | 1.08 | 1.19 | $1.15_{\pm0.04}$ |
| | ALL | 0.95 | 0.91 | 1.11 | 1.06 | 1.09 | $1.01_{\pm0.08}$ |
| | DMPO | 1.13 | 1.18 | 1.34 | 1.10 | 1.21 | $1.17_{\pm0.06}$ |
| SELT 20% | PD (high) | 0.53 | 0.46 | 0.47 | 0.54 | 0.41 | $0.48_{\pm0.05}$ |
| | PD (mid) | 1.05 | 1.08 | 1.16 | 1.15 | 1.16 | $1.12_{\pm0.05}$ |
| | PD (nl) | 1.15 | 1.06 | 1.16 | 1.16 | 1.15 | $1.13_{\pm0.04}$ |
| | RAND | 0.96 | 0.91 | 1.11 | 1.16 | 1.00 | $1.01_{\pm0.09}$ |
| | RAF | 1.10 | 1.06 | 1.36 | 1.16 | 1.21 | $1.15_{\pm0.08}$ |
| | PD (rati.) | 1.08 | 1.11 | 1.23 | 1.18 | 1.26 | $1.17_{\pm0.07}$ |
| | PD (ours) | 1.15 | 1.13 | 1.24 | 1.07 | 1.27 | $1.17_{\pm0.07}$ |
| SELT 30% | PD (high) | 0.61 | 0.53 | 0.57 | 0.74 | 0.53 | $0.59_{\pm0.08}$ |
| | PD (mid) | 1.11 | 1.13 | 1.26 | 1.15 | 1.22 | $1.17_{\pm0.05}$ |
| | PD (nl) | 1.17 | 1.15 | 1.23 | 1.09 | 1.25 | $1.18_{\pm0.06}$ |
| | RAND | 0.97 | 1.03 | 1.03 | 1.06 | 1.05 | $1.03_{\pm0.03}$ |
| | RAF | 1.13 | 1.19 | 1.27 | 1.14 | 1.23 | $1.19_{\pm0.05}$ |
| | PD (rati.) | 1.16 | 1.14 | 1.29 | 1.22 | 1.29 | $1.21_{\pm0.06}$ |
| | PD (ours) | 1.16 | 1.18 | 1.39 | 1.19 | 1.30 | $1.23_{\pm0.07}$ |
| SELT 40% | PD (high) | 0.67 | 0.66 | 0.73 | 0.78 | 0.55 | $0.66_{\pm0.08}$ |
| | PD (mid) | 1.11 | 1.23 | 1.27 | 1.23 | 1.24 | $1.21_{\pm0.05}$ |
| | PD (nl) | 1.20 | 1.23 | 1.37 | 1.12 | 1.24 | $1.22_{\pm0.06}$ |
| | RAND | 1.01 | 1.01 | 1.16 | 1.10 | 1.11 | $1.07_{\pm0.05}$ |
| | RAF | 1.13 | 1.19 | 1.30 | 1.17 | 1.26 | $1.20_{\pm0.05}$ |
| | PD (rati.) | 1.16 | 1.18 | 1.33 | 1.19 | 1.27 | $1.22_{\pm0.05}$ |
| | PD (ours) | 1.13 | 1.20 | 1.22 | 1.23 | 1.27 | $1.21_{\pm0.05}$ |
| SELT 50% | PD (high) | 0.84 | 0.76 | 0.85 | 0.88 | 0.83 | $0.83_{\pm0.04}$ |
| | PD (mid) | 1.15 | 1.16 | 1.33 | 1.16 | 1.26 | $1.20_{\pm0.06}$ |
| | PD (nl) | 1.11 | 1.27 | 1.33 | 1.23 | 1.21 | $1.22_{\pm0.06}$ |
| | RAND | 1.03 | 1.03 | 1.09 | 1.05 | 1.06 | $1.05_{\pm0.02}$ |
| | RAF | 1.19 | 1.32 | 1.31 | 1.17 | 1.24 | $1.24_{\pm0.06}$ |
| | PD (rati.) | 1.19 | 1.30 | 1.28 | 1.18 | 1.31 | $1.26_{\pm0.06}$ |
| | PD (ours) | 1.20 | 1.22 | 1.28 | 1.18 | 1.30 | $1.24_{\pm0.04}$ |

Table E.7: Detailed win score across five test datasets of Llama model on 20% conflict level dataset.

| | Method | WizardLM | Sinstruct | Vicuna | Koala | Lima | AW |
|---|---|---|---|---|---|---|---|
| FULL | SFT | 0.69 | 0.62 | 0.63 | 0.71 | 0.60 | $0.64_{\pm0.05}$ |
| | OVA. | 1.00 | 1.00 | 1.00 | 1.00 | 1.00 | $1.00_{\pm0.00}$ |
| | AVG. | 1.15 | 1.16 | 1.17 | 1.08 | 1.19 | $1.15_{\pm0.04}$ |
| | ALL | 0.92 | 0.89 | 0.90 | 0.91 | 0.93 | $0.91_{\pm0.02}$ |
| | DMPO | 0.99 | 1.09 | 1.28 | 1.02 | 1.18 | $1.09_{\pm0.09}$ |
| SELT 20% | PD (high) | 0.57 | 0.38 | 0.36 | 0.54 | 0.38 | $0.45_{\pm0.09}$ |
| | PD (mid) | 1.05 | 1.00 | 1.32 | 1.14 | 1.07 | $1.08_{\pm0.08}$ |
| | PD (nl) | 1.09 | 1.04 | 1.36 | 1.14 | 1.12 | $1.12_{\pm0.08}$ |
| | RAND | 0.98 | 0.85 | 1.01 | 0.98 | 0.99 | $0.95_{\pm0.06}$ |
| | RAF | 1.12 | 1.04 | 1.25 | 1.12 | 1.16 | $1.12_{\pm0.06}$ |
| | PD (rati.) | 1.13 | 1.12 | 1.39 | 1.12 | 1.15 | $1.15_{\pm0.07}$ |
| | PD (ours) | 1.19 | 1.04 | 1.34 | 1.16 | 1.23 | $1.17_{\pm0.09}$ |
| SELT 30% | PD (high) | 0.54 | 0.43 | 0.43 | 0.59 | 0.39 | $0.47_{\pm0.07}$ |
| | PD (mid) | 1.08 | 1.13 | 1.19 | 1.21 | 1.21 | $1.16_{\pm0.05}$ |
| | PD (nl) | 1.11 | 1.20 | 1.32 | 1.14 | 1.23 | $1.19_{\pm0.06}$ |
| | RAND | 0.97 | 0.94 | 1.04 | 1.06 | 1.02 | $1.00_{\pm0.05}$ |
| | RAF | 1.08 | 1.17 | 1.33 | 1.14 | 1.19 | $1.17_{\pm0.06}$ |
| | PD (rati.) | 1.17 | 1.22 | 1.31 | 1.17 | 1.28 | $1.23_{\pm0.05}$ |
| | PD (ours) | 1.19 | 1.23 | 1.30 | 1.16 | 1.22 | $1.21_{\pm0.04}$ |
| SELT 40% | PD (high) | 0.58 | 0.51 | 0.56 | 0.69 | 0.51 | $0.56_{\pm0.07}$ |
| | PD (mid) | 1.09 | 1.15 | 1.29 | 1.19 | 1.22 | $1.18_{\pm0.06}$ |
| | PD (nl) | 1.19 | 1.20 | 1.34 | 1.13 | 1.29 | $1.22_{\pm0.07}$ |
| | RAND | 0.99 | 0.95 | 1.15 | 1.03 | 1.07 | $1.02_{\pm0.06}$ |
| | RAF | 1.12 | 1.16 | 1.33 | 1.19 | 1.28 | $1.20_{\pm0.07}$ |
| | PD (rati.) | 1.19 | 1.20 | 1.39 | 1.15 | 1.24 | $1.22_{\pm0.06}$ |
| | PD (ours) | 1.17 | 1.24 | 1.33 | 1.23 | 1.24 | $1.23_{\pm0.04}$ |
| SELT 50% | PD (high) | 0.70 | 0.57 | 0.77 | 0.73 | 0.72 | $0.68_{\pm0.07}$ |
| | PD (mid) | 1.19 | 1.17 | 1.23 | 1.25 | 1.22 | $1.21_{\pm0.03}$ |
| | PD (nl) | 1.15 | 1.24 | 1.30 | 1.20 | 1.22 | $1.21_{\pm0.04}$ |
| | RAND | 1.04 | 1.06 | 1.13 | 1.09 | 1.05 | $1.06_{\pm0.02}$ |
| | RAF | 1.20 | 1.24 | 1.38 | 1.18 | 1.29 | $1.25_{\pm0.06}$ |
| | PD (rati.) | 1.20 | 1.26 | 1.41 | 1.11 | 1.27 | $1.23_{\pm0.08}$ |
| | PD (ours) | 1.16 | 1.27 | 1.29 | 1.19 | 1.25 | $1.23_{\pm0.05}$ |

Table E.8: Detailed win score across five test datasets of Llama model on 30% conflict level dataset.

| | Method | WizardLM | Sinstruct | Vicuna | Koala | Lima | AW |
|---|---|---|---|---|---|---|---|
| FULL | SFT | 0.81 | 0.84 | 0.71 | 0.66 | 0.73 | $0.76_{\pm0.07}$ |
| | OVA. | 1.00 | 1.00 | 1.00 | 1.00 | 1.00 | $1.00_{\pm0.00}$ |
| | AVG. | 0.99 | 1.04 | 0.91 | 1.02 | 1.10 | $1.03_{\pm0.05}$ |
| | ALL | 1.05 | 1.07 | 0.87 | 1.01 | 1.01 | $1.02_{\pm0.05}$ |
| | DMPO | 1.05 | 1.22 | 1.02 | 0.98 | 1.09 | $1.09_{\pm0.09}$ |
| SELT 20% | RAND | 0.93 | 1.02 | 0.96 | 0.89 | 0.85 | $0.92_{\pm0.07}$ |
| | RAF | 0.96 | 0.99 | 0.91 | 0.88 | 0.91 | $0.94_{\pm0.04}$ |
| | PD (rati.) | 0.93 | 1.14 | 0.81 | 0.98 | 0.98 | $0.99_{\pm0.09}$ |
| | PD (ours) | 1.01 | 1.14 | 0.96 | 0.92 | 1.03 | $1.03_{\pm0.07}$ |
| SELT 30% | RAND | 0.94 | 1.01 | 0.91 | 0.89 | 0.93 | $0.94_{\pm0.04}$ |
| | RAF | 1.00 | 1.10 | 0.98 | 0.97 | 0.98 | $1.01_{\pm0.05}$ |
| | PD (rati.) | 1.06 | 1.12 | 1.08 | 0.88 | 0.98 | $1.02_{\pm0.08}$ |
| | PD (ours) | 1.03 | 1.27 | 0.94 | 1.08 | 1.05 | $1.10_{\pm0.10}$ |
| SELT 40% | RAND | 0.95 | 0.98 | 0.96 | 0.91 | 0.89 | $0.94_{\pm0.04}$ |
| | RAF | 1.12 | 1.15 | 1.02 | 1.06 | 1.12 | $1.11_{\pm0.04}$ |
| | PD (rati.) | 1.12 | 1.21 | 1.01 | 1.05 | 1.16 | $1.13_{\pm0.06}$ |
| | PD (ours) | 1.10 | 1.25 | 1.05 | 1.17 | 1.18 | $1.17_{\pm0.06}$ |
| SELT 50% | RAND | 0.98 | 1.07 | 0.81 | 1.00 | 0.95 | $0.98_{\pm0.07}$ |
| | RAF | 1.10 | 1.18 | 1.11 | 1.06 | 1.05 | $1.10_{\pm0.05}$ |
| | PD (rati.) | 1.07 | 1.19 | 1.06 | 1.14 | 1.13 | $1.13_{\pm0.04}$ |
| | PD (ours) | 1.14 | 1.24 | 1.16 | 1.19 | 1.18 | $1.19_{\pm0.04}$ |

Table E.9: Detailed win score across five test datasets of Llama model on HelpSteer.

| | Method | WizardLM | Sinstruct | Vicuna | Koala | Lima | AW |
|---|---|---|---|---|---|---|---|
| FULL | SFT | 0.78 | 0.78 | 0.74 | 0.85 | 0.71 | 0.77±0.05 |
| | OVA. | 1.00 | 1.00 | 1.00 | 1.00 | 1.00 | 1.00±0.00 |
| | AVG. | 1.13 | 1.19 | 1.24 | 1.12 | 1.31 | 1.20±0.08 |
| | ALL | 1.06 | 1.13 | 1.17 | 1.10 | 1.25 | 1.15±0.07 |
| | DMPO | 1.24 | 1.32 | 1.32 | 1.37 | 1.37 | 1.33±0.05 |
| SELT 20% | RAND | 1.08 | 1.06 | 1.14 | 1.18 | 1.15 | 1.12±0.04 |
| | RAF | 1.17 | 1.11 | 1.23 | 1.15 | 1.22 | 1.17±0.04 |
| | PD (rati.) | 1.15 | 1.13 | 1.17 | 1.18 | 1.19 | 1.16±0.02 |
| | PD (ours) | 1.23 | 1.21 | 1.34 | 1.29 | 1.30 | 1.26±0.04 |
| SELT 30% | RAND | 1.10 | 1.07 | 1.11 | 1.17 | 1.20 | 1.14±0.05 |
| | RAF | 1.20 | 1.15 | 1.28 | 1.20 | 1.37 | 1.24±0.09 |
| | PD (rati.) | 1.13 | 1.16 | 1.24 | 1.20 | 1.27 | 1.20±0.05 |
| | PD (ours) | 1.22 | 1.24 | 1.29 | 1.23 | 1.34 | 1.27±0.05 |
| SELT 40% | RAND | 1.16 | 1.11 | 1.13 | 1.16 | 1.20 | 1.15±0.03 |
| | RAF | 1.25 | 1.16 | 1.29 | 1.31 | 1.37 | 1.28±0.08 |
| | PD (rati.) | 1.16 | 1.20 | 1.24 | 1.27 | 1.25 | 1.22±0.04 |
| | PD (ours) | 1.23 | 1.22 | 1.43 | 1.32 | 1.36 | 1.30±0.07 |
| SELT 50% | RAND | 1.16 | 1.07 | 1.26 | 1.18 | 1.18 | 1.15±0.05 |
| | RAF | 1.25 | 1.27 | 1.32 | 1.36 | 1.37 | 1.32±0.05 |
| | PD (rati.) | 1.18 | 1.20 | 1.28 | 1.23 | 1.25 | 1.22±0.03 |
| | PD (ours) | 1.21 | 1.22 | 1.35 | 1.42 | 1.45 | 1.33±0.11 |

Table E.10: Detailed win score across five test datasets of Qwen model on UltraFeedback.

| | Method | WizardLM | Sinstruct | Vicuna | Koala | Lima | AW |
|---|---|---|---|---|---|---|---|
| FULL | SFT | 0.78 | 0.78 | 0.74 | 0.85 | 0.71 | 0.77±0.05 |
| | OVA. | 1.00 | 1.00 | 1.00 | 1.00 | 1.00 | 1.00±0.00 |
| | AVG. | 1.00 | 1.04 | 1.06 | 1.12 | 0.99 | 1.03±0.05 |
| | ALL | 1.03 | 1.07 | 1.10 | 0.96 | 1.01 | 1.03±0.04 |
| | DMPO | 1.06 | 1.16 | 1.00 | 1.09 | 1.06 | 1.08±0.05 |
| SELT 20% | RAND | 0.97 | 0.97 | 0.84 | 0.86 | 0.89 | 0.92±0.05 |
| | RAF | 0.93 | 0.96 | 0.84 | 0.88 | 0.95 | 0.93±0.04 |
| | PD (rati.) | 1.02 | 1.10 | 1.04 | 1.02 | 0.99 | 1.03±0.04 |
| | PD (ours) | 0.97 | 1.11 | 0.96 | 0.97 | 1.07 | 1.03±0.06 |
| SELT 30% | RAND | 0.95 | 0.94 | 1.09 | 0.92 | 0.95 | 0.95±0.04 |
| | RAF | 0.95 | 1.02 | 1.03 | 1.01 | 0.98 | 0.99±0.03 |
| | PD (rati.) | 1.04 | 1.13 | 1.08 | 1.19 | 1.08 | 1.10±0.05 |
| | PD (ours) | 1.02 | 1.21 | 1.11 | 1.08 | 1.09 | 1.10±0.07 |
| SELT 40% | RAND | 1.02 | 0.97 | 0.88 | 0.96 | 0.93 | 0.96±0.04 |
| | RAF | 1.02 | 1.15 | 0.93 | 1.01 | 1.00 | 1.04±0.07 |
| | PD (rati.) | 1.07 | 1.25 | 1.17 | 1.11 | 1.17 | 1.16±0.06 |
| | PD (ours) | 1.13 | 1.22 | 1.11 | 1.09 | 1.19 | 1.16±0.05 |
| SELT 50% | RAND | 0.99 | 0.98 | 0.90 | 1.02 | 0.97 | 0.98±0.03 |
| | RAF | 1.08 | 1.03 | 1.11 | 1.08 | 1.00 | 1.05±0.04 |
| | PD (rati.) | 1.15 | 1.27 | 1.06 | 1.29 | 1.20 | 1.21±0.07 |
| | PD (ours) | 1.11 | 1.35 | 1.16 | 1.15 | 1.26 | 1.22±0.09 |

Table E.11: Detailed win score across five test datasets of Qwen model on HelpSteer.

