# OpenReview forum: "Data Selection for LLM Alignment Using Fine-Grained Preferences"
_ICLR.cc/2026/Conference — ICLR 2026 Poster_

### Official Review · Reviewer_xH9J · 2025-10-28

**Soundness:** 3
**Presentation:** 3
**Contribution:** 3
**Rating:** 6
**Confidence:** 4

**Summary:**

This paper addresses a critical challenge in aligning Large Language Models (LLMs): effectively utilizing datasets composed of multiple, fine-grained human preferences (e.g., helpfulness, honesty) which often contain inherent conflicts and noise. The authors identify that standard alignment methods like DPO struggle with such aggregated data. Their core contribution is a data-centric approach that reframes the problem as data selection. They first formulate a Direct Fine-Grained Preference Optimization (DFPO) objective, which introduces a Preference Divergence (PD) term that quantifies the conflict between a specific fine-grained preference and the consensus of all others. Instead of directly optimizing this complex objective, they theoretically and empirically demonstrate that selecting a subset of data with the most negative PD values (indicating high consensus across aspects) for standard DPO training leads to superior performance. The proposed method involves training small, de-biased proxy reward models for each sub-preference to estimate the PD terms. Extensive experiments on datasets derived from UltraFeedback and HelpSteer show that their method consistently outperforms full-data alignment and other baselines, achieving better results using only 30-50% of the data, while also being more robust to increasing levels of preference conflict.

**Strengths:**

1. he paper clearly identifies a significant and under-explored problem—handling noise and conflicts in aggregated fine-grained preference data. The motivation is strong, backed by a statistical analysis of a real-world dataset (UltraFeedback) showing a high rate of preference conflicts.

2. The progression from the DFPO formulation to the PD term, and then to the data selection strategy, is elegant and well-justified. The theoretical analysis (Theorems 3.4 & 3.5) provides a solid foundation for the simple yet effective heuristic of selecting samples with the most negative PD.

3. The experimental section is thorough and convincing. The authors demonstrate:

**Weaknesses:**

1. The entire selection process hinges on the quality of the proxy reward models. While the paper shows low sensitivity to the model's size and training data amount, it does not deeply explore scenarios where these proxy models might fail catastrophically (e.g., on out-of-distribution data or aspects that are inherently difficult to model). The method inherits all the challenges of reward modeling.

2.  The method implicitly assumes all sub-preferences are equally important by summing their pseudo-reward gaps uniformly in the PD term. In reality, some aspects (e.g., "truthfulness") might be more critical than others (e.g., "verbosity"). The work does not explore weighted PD terms or user-specified preference hierarchies.

3. The experiments are conducted on models up to 8B parameters. It is unclear how the method would scale to very large models (e.g., 70B+), where the computational savings would be even more dramatic but the dynamics of fine-tuning can differ. Furthermore, the evaluation, while comprehensive, is primarily based on general-purpose chat benchmarks; testing on more specific safety-critical or instruction-following benchmarks would strengthen the claims about robust alignment.

4. The derivation of the DFPO objective in the main text (and Appendix A.1.1) is quite dense and could be challenging for a reader to follow. A more intuitive, step-by-step explanation of how the PD term emerges from the multi-aspect RLHF objective would improve accessibility.

**Questions:**

1. How would the method perform if the fine-grained aspects were not orthogonal or even negatively correlated? For instance, if "conciseness" and "completeness" were both aspects, the definition of "conflict" might need refinement.

2. The length bias mitigation is a crucial component. Was any analysis done to determine the optimal value for the length penalty coefficient ρ? Is it robust across different datasets, or does it require tuning?

3. The paper focuses on selecting data for a single round of DPO training. Could this PD-based selection be integrated into an iterative training loop, where the aligned model is used to re-estimate rewards and re-select data for further training?

4. The "PD (rati.)" baseline, which uses ground-truth ratings, performs comparably well in some cases. Does this suggest that with perfectly reliable, continuous ratings (instead of binary preferences), the need for proxy models could be eliminated? What are the relative trade-offs between the annotation cost of ratings vs. the computational cost of training proxy models?

---

> ### Author Response · Authors · 2025-11-18
> **Author Response to Reviewer xH9J**
>
> Dear Reviewer xH9J:
>
> Thank you for the valuable feedback on our paper. We appreciate the time and effort you have put into reviewing our work. We have carefully read your review and addressed your concerns as follows.
>
> **Q1: What if the fine-grained aspects were not orthogonal or even negatively correlated?**
>
> A1: Thank you for this insightful question, which helps us precisely define our work's scope. Our paper (§1, §3.1) deals with multiple compatible fine-grained aspects. Our definition of conflict is at the data level: it refers to a specific sample that cannot excel on all aspects simultaneously due to quality issues . Our PD method filters data with high conflict (low quality), selecting a subset whose implicit preference is closer to the overall ground-truth. We have strengthened our definition of conflict to avoid this potential ambiguity (§3.1).
>
> We agree that correlation of different aspects can exist in some scenarios, which we categorize into two cases:
>
> 1. **Positive correlation:** E.g., "helpfulness" and "usefulness". This could over-weight similar aspects in the current PD calculation. A valuable extension is to quantify this correlation and down-weight redundant aspects in the PD estimation for a more precise measure.
> 2. **Negative correlation:** E.g., "conciseness" vs. "completeness". These fine-grained criteria are definitionally incompatible and in conflict, meaning a response that excels in all aspects could not exist. This fundamentally changes the problem to one of Pareto optimization, which is outside our current scope of seeking a single, best-performing model.
>
> In summary, our work can be naturally extended for positive dependencies by introducing correlation weights to mitigate redundancy. The case of negative dependencies represents a different but important research direction, as further discussed in Appendix §D.
>
> **Q2: Analysis on the coefficient** $\rho$
>
> A2: We have added more detailed ablation experiments on hyperparameters in §5.6. They confirm that the performance is stable when $\rho$ is set within a reasonable and moderate range. A suitable and fixed choice of 1e-3 works well across all the experiments we conducted, and carefully tuning for each specific dataset is not required.
>
> **Q3: Integrate the method into an iterative training loop.**
>
> A3: Thank you for this insightful question, which provides an important guiding direction for future work. Our current work focuses on establishing the foundation for data selection in the standard offline DPO pipeline. Integrating PD selection into an iterative loop is a natural and highly promising next step. While there are practical implementation gaps to bridge, such as how to efficiently co-update the proxy reward models with the policy to filter newly collected data in each round, we believe this is a critical area for active exploration.  We have also added a related discussion in Appendix §D.
>
> **Q4: Discussion on the trade-off between the annotation cost of ratings vs. the computational cost of training proxy models.**
>
> A4: This is an excellent observation. The strong performance of PD (rati.)  serves as a powerful validation of our core theory: that filtering based on PD term is highly effective.
>
> We agree with your point that if one could obtain fully reliable ratings for every sample across all aspects, one could (to an extent) bypass the need for proxy models. However, this approach suffers from an extremely high annotation cost. It requires an annotator to provide a fine-grained, accurate rating for all aspects on every single data point. This is not just K times the cost of our method, but far more, as K ratings are much harder than one binary annotation. In contrast, we only require a single, simple, binary preference on one sub-aspect per sample.
>
> Our method exchanges a small, one-time training cost (training small proxy RMs) for a massive saving in annotation cost. While real ratings were invaluable for validating our method, our method is far more economical and practical for real-world applications. That said, if a fine-grained aspect proves too complex for a proxy RM to model effectively, the real rating annotation method still retains its value.

---

> > ### Author Response · Authors · 2025-11-18
> > **Author Response to Reviewer xH9J**
> >
> > **W1: The method inherits all the challenges of reward modeling.**
> >
> > R1: We agree that our method relies on the proxy reward models and thus inherits the general challenges of reward modeling. Our effort to mitigate this is conceptual: we argue it is fundamentally easier to model a simple, clear, fine-grained preference than a complex, mixed overall preference. By decomposing the task, we simplify the criteria for each proxy reward modeling, which helps to *mitigate* (though does not eliminate) the learning difficulty.
> >
> > Additionally, our contribution focuses on the data selection methodology to mitigate data issues. Building perfectly robust RMs (e.g., against OOD data) is a broader and more profound challenge, but it is outside our current scope. We have explicitly added a discussion of these potential risks of reward modeling to Appendix §D: Limitation and Future Work.
> >
> > **W2: Implicit assumption that all sub-preferences are equally important.**
> >
> > R2: This is a very important and accurate observation. This assumption stems from our theoretical starting point, Eq. (1), where we chose to optimize the standard, non-weighted average reward.
> > As we also discussed in our response to Reviewer KGaJ (Q2), a natural and valuable extension is to change this starting point to a weighted average reward, which assigns different priorities to different aspects. This would naturally lead to a weighted or user-specified preference hierarchy. We fully agree that exploring this is a critical and immediate next step, as we have already added and highlighted this in Appendix §D.
> >
> > **W3: Concerns on model scale and types of benchmark.**
> >
> > R3: Thank you for these important points.
> >
> > 1) We acknowledge this limit is due to our computational constraints. We would like to clarify that our work is fundamentally an investigation into data quality, which is a model-agnostic challenge. We strongly agree with the reviewer's insight that the cost-saving benefits of data selection would be even more significant in very large models.
> >
> > 2) We agree that validating on specific domains is crucial. However, many existing specific benchmarks do not meet our core experimental requirement: they often lack the multiple fine-grained aspect annotations (especially K>2). This is also the rationale for our current dataset choice, as we have now detailed in Appendix §C.2.
> >
> > 3) We have added new results on our proprietary real-world downstream application in §5.7, for which we collected data using 4 fine-grained aspects. Our method showed consistent improvements, strengthening our claims when applied to more specialized domains and tasks.
> >
> > **W4: A more intuitive, step-by-step explanation of PD and DFPO.**
> >
> > R4: Thank you for the suggestion. We have added further explanations to the derivation in Appendix §A.1.1 to help improve comprehension.

---

> ### Author Response · Authors · 2025-11-25
> **Acknowledgement to Reviewer xH9J**
>
> We sincerely thank you for your positive feedback and for raising the score. We deeply appreciate the time and effort you dedicated to reviewing our paper and rebuttal; your constructive comments have been instrumental in improving the quality of our work.

---

> > ### Comment · Reviewer_xH9J · 2025-11-25
> >
> > I have acknowledged the rebuttal

---

### Official Review · Reviewer_qeXk · 2025-10-31

**Soundness:** 4
**Presentation:** 2
**Contribution:** 3
**Rating:** 8
**Confidence:** 3

**Summary:**

The paper reframes preference learning under mixed and potentially conflicting preferences as the selection for a subset of training data whose implicit preference is closest to the ground-truth preference. A data-selection algorithm (DFPO) is accordingly proposed and evaluated extensively.

**Strengths:**

1. Detailed problem formulation and the novel transformation of the problem into data selection methods instead of algorithm development is interesting.

2. Empirical coverage is thorough.

**Weaknesses:**

1. With ever-larger models and compute, scaling-laws may simply “wash out” moderate preference noise; the urgency of the problem is not demonstrated.

2. The method operates on a fixed dataset and is demonstrated only with the now “classical” DPO pipeline.  Readers working with on-policy RL extensions are unlikely to see an immediate hook.  Extending DFPO to iterative regimes like iterative DPO would greatly widen its appeal.

3. If a dataset contains several conflicting preferences, DFPO appears to return the largest conflict-free subset.  When preferences are equally strong and mutually incompatible, is the algorithm simply selecting the majority preference?  A crisp explanation that shows what is really done by DFPO is missing.

4. Many experiments are presented, but each is analysed briefly. Spare more pages to careful analysis will be helpful.

**Questions:**

1. The formalism treats (human) preference as an abstract concept, while the same conflict could arise when mixing code, math, etc.  How should the “ground-truth preference” be defined in such multi-task settings where no single human ranking exists?

---

> ### Author Response · Authors · 2025-11-18
> **Author Response to Reviewer qeXk**
>
> Dear Reviewer qeXk:
>
> Thank you for the valuable feedback on our paper. We appreciate the time and effort you have put into reviewing our work. We have carefully read your review and addressed your concerns as follows.
>
> **Q1: Discussion on  "ground-truth preference" in multi-task settings (e.g., code, math) where no single human ranking exists?**
>
> A1: Thank you for this profound question. Our current work (as described in §1)  focuses on a more specific scenario: a task with an overall preference, which can be decomposed into compatible fine-grained components.
>
> We completely agree with your view: in a complex multi-task setting, a single "ground-truth preference" may not exist. The optimal standards for different tasks can themselves be conflicting, leading to conflicts even at the fine-grained level. This, as you may imply, becomes a Pareto optimization problem, which is a different but very important problem from our current work. Therefore, we believe combining effective data selection with solving for the Pareto frontier in such multi-task scenarios is an important and interesting future direction, which we have added discussion in Appendix §D.
>
> **W1: The urgency of the data selection problem is not demonstrated.**
>
> R1: Thank you for raising this important point. Scaling laws are indeed powerful, especially in pre-training on massive datasets with moderate noise. However, a growing body of research [1-5], as we also discussed in our related work (§2.2), shows that data quality is critical for both pre-training and post-training. This is especially true for post-training, which often involves much smaller datasets, making data quality paramount. The results in our study also validate this (§5.3): increased data noise and conflicts cause the performance of full-data alignment to degrade drastically. Our effective selection method mitigates this performance drop, demonstrating the necessity of the active and effective data curation.
>
> We have added a more explicit discussion on the necessity of data selection for LLM in §2.2 to better highlight the problem's urgency.
>
> [1] Data Selection for Language Models via Importance Resampling. NeurIPS 2023.
>
> [2] LESS: Selecting Influential Data for Targeted Instruction Tuning. ICML 2024.
>
> [3] From Quantity to Quality: Boosting LLM Performance with Self-Guided Data Selection for Instruction Tuning. NAACL 2024.
>
> [4] s1: Simple Test-Time Scaling. EMNLP 2025.
>
> [5] Data Selection via Optimal Control for Language Models. ICLR 2025.
>
> **W2: Extending to iterative regimes.**
>
> R2: We agree this is a valuable extension. Our current work focuses on establishing the data selection methodology in the standard offline DPO pipeline.  However, we are also interested in combining our work with other paradigms. For example: 1) In Iterative DPO, the proxy reward models used for PD estimation could be updated alongside the policy model to filter newly collected preference data in each round. 2) In On-policy RL, this multi-aspect PD evaluation could be used to filter or weight higher-quality experiences, replacing or augmenting a single scalar reward signal. While there are some technical and practical gaps to bridge, we believe extending our work to iterative and on-policy paradigms is a natural and important future direction. We have also added a related discussion in Appendix §D.
>
> **W3: Does the algorithm just select the majority when aspects are mutually incompatible?**
>
> R3: Thank you for this crucial question, which allows us to clarify what our work is really doing. 1) Annotation Conflict: We assume a setting with compatible aspects (e.g., "helpfulness" and "honesty," which are both desirable). Our introduced conflict refers to the sample-level annotation issue, where a specific sample fails to excel in all aspects simultaneously due to quality issues. 2) Definitional Incompatibility: Mutually incompatible objectives (e.g., "conciseness" vs. "completeness").
>
> Our work focuses on the first case. As you perfectly summarized in your review, our goal is to find a subset where the "implicit preference is closest to the ground-truth," not to choose among incompatible objectives. The case of definitionally incompatible aspects is a different, valid problem that falls under Pareto optimization, which is outside our current scope and an important area for our future research.
>
> **W4: Need more pages for careful analysis of experiments.**
>
> R4: We thank the reviewer for this feedback. We acknowledge that due to ICLR's strict page limits, we had to highly compress our main analysis in the paper. We have already added a more detailed analysis of all experimental results in our manuscript.

---

### Official Review · Reviewer_vgsY · 2025-11-01

**Soundness:** 3
**Presentation:** 3
**Contribution:** 3
**Rating:** 4
**Confidence:** 3

**Summary:**

This paper addresses LLM alignment with human preferences, emphasizing the difficulties arising when using fine-grained, aspect-specific preferences rather than singular “overall” preference labels. The core proposal is a data-centric, preference divergence (PD)-based selection strategy that identifies and utilizes the most reliable samples from aggregated, potentially conflicting, fine-grained preference datasets.
The authors provide theoretical grounding for their selection method (establishing loss-bound optimality), introduce approaches to estimate PD in practice (including bias mitigation), and conduct comprehensive experiments demonstrating that selective alignment using just a fraction (e.g., 30\%) of filtered data can outperform traditional full-dataset alignment in both performance and computational efficiency.

**Strengths:**

1. The authors clearly demonstrate motivation for the problem, where aggregating fine-grained preferences introduces conflicts, redundancy, and noise that degrade LLM alignment.
2. The development of loss bounds and the selection optimality result underpin the proposed data selection strategy with rigorous analysis, providing compelling mathematical justification for selecting samples by most-negative PD.
3. Extensive evaluation: The method is thoroughly evaluated against full-data and alternative selection baselines, across datasets (UltraFeedback, HelpSteer), models (Llama, Qwen), and varying conflict levels. PD-based selection (especially “ours”) consistently outperforms others—even with substantial data reduction—while being robust to proxy model choice and ablation.

**Weaknesses:**

1. I am not familiar with this research scope, but the current evaluation focuses on UltraFeedback and HelpSteer, and their derived conflict settings are limited. The author should conduct experiments with more advanced benchmarks for a clear demonstration of their effectiveness.
2.  The empirical studies do not report in-depth on the sensitivity of the method to hyperparameters (e.g., $\lambda$, quantile level $\gamma$, length penalty $\rho$, sampling ratio $p_r$), aside from the generic selection budget. Although performance seems stable as shown in Figures 3 and 4, the rationale for particular choices and the potential for overfitting (especially when “tuning” for the best selection cut-off) are not thoroughly analyzed.

**Questions:**

See weaknesses.

---

> ### Author Response · Authors · 2025-11-18
> **Author Response to Reviewer vgsY**
>
> Dear Reviewer vgsY:
>
> Thank you for the valuable feedback on our paper. We appreciate the time and effort you have put into reviewing our work. We have carefully read your review and addressed your concerns as follows.
>
> **Q1: Rationale on the benchmarks used and limited conflict settings.**
>
> A1: Thank you for this valuable feedback. We fully agree that validating our method on diverse and advanced benchmarks is crucial.
>
> 1. **(Rationale of the dataset used)** Our work (as described in §1 and §3.1)  specifically targets the challenge of aligning LLMs with aggregated datasets of multiple fine-grained preferences, and the inherent noise and conflicts that arise.  Therefore, to rigorously evaluate, we require large-scale datasets that provide annotations for multiple aspects (especially K>2). To our knowledge, UltraFeedback (4 aspects ) and HelpSteer (5 aspects) are the primary large-scale public datasets that meet this specific requirement. We have also added a detailed rationale of the dataset used in Appendix §C.2.
> 2. (**Response to limited conflict settings**) We also wish to clarify that our derived conflict settings were a rigorous controlled experiment designed to explicitly validate the robustness and effectiveness of our method against varying conflict levels, as discussed in §5.3.
>
>     More importantly, our method addresses data quality holistically, not just in conflict scenarios. Non-conflicting samples can still vary significantly in quality (e.g., ambiguous or overly difficult samples, see §5.2). Empirically, our method (using the top 30% of data) outperforms the AVG. baseline (which features better preference consistency) on the dataset with low (10%) conflict levels, confirming that quality disparities persist even in the non-conflicting samples. Our selection mechanism (via the PD term) effectively targets the highest-quality data, simultaneously excluding both conflicting samples and non-conflicting but low-value samples to maximize training efficiency.
>
>     Therefore, while conflicts are a common issue in aggregated fine-grained datasets, we argue that simulating extreme conflict settings is not necessary for demonstrating real-world applicability. The core value of this study lies in the ability to distill the most effective training subset from the original dataset.
> 3. **(Additional evidence)** We have also added results on our proprietary real-world downstream application (§5.7) by collecting a dataset using 4 fine-grained preferences, which showed consistent improvements. This helps validate the effectiveness and practicality of our approach, even when applied to more specialized domains and tasks.
>
> **Q2: Sensitivity analysis of other hyperparameters and the potential for overfitting the selection cut-off.**
>
> A2: Thank you for this feedback. We address your points in two parts:
>
> 1. Hyperparameters:
>     1. For $p_r$: Figure 5 demonstrates our method's low sensitivity to the proxy model's training data amount. We chose 30% as a balance of cost and performance.  We have also included a more detailed analysis in the corresponding section.
>     2. For $\gamma$, and $\rho$: We have added supplementary ablation experiments in §5.6. They confirm our method is stable in a moderate range and only degrades at extremes, showing careful tuning is not required. Therefore, we adopted a fixed setting across all other experiments.
> 2. Selection Budget ($\lambda$):
>
>     Results in Figures 3, 4, C.2 and C.3 demonstrate robustness, not overfitting. PD selection strategy based on either our estimation or fine-grained ratings provided by datasets is consistently superior across the entire 20%-50% range tested. Thus, the $\lambda$ in our main experiments is an efficiency-performance trade-off, not a cherry-picked point.
>
> We once again thank you for your valuable feedback, based on which we have refined the manuscript to enhance its quality. We look forward to further discussion.

---

### Official Review · Reviewer_KGaJ · 2025-11-09

**Soundness:** 3
**Presentation:** 3
**Contribution:** 3
**Rating:** 6
**Confidence:** 4

**Summary:**

The authors proposed a data-centric approach to align LLMs through the effective use of fine-grained preferences. They studied a data selection problem and propose an effective strategy to identify a subset of data corresponding to the most negative PD values for the training.

**Strengths:**

A balanced sampling strategy is applied to mitigate the intrinsic bias towards longer responses that are favored regardless of quality.

A penalty term is introduced into the reward model to discourage length bias as well.

The paper is well written and easy to follow.

**Weaknesses:**

See the below questions.

**Questions:**

To address the practical challenge in DFPO, i.e., the high computational cost and the risk of instability from unavailable/unreliable reward models, the authors filter the dataset to construct a high-quality subset for the training. Besides the data selection strategy, are there other strategies to address the challenge? Is it possible to apply LLMs’ distillation techniques?

The authors extend the principle of DPO to the fine-grained preference alignment setting by introducing an additional term (equation 4), i.e., the preference divergence one. Are there other way to define the objective for the purpose? Any discussions?

The authors consider K fine-grained criteria. What if these K fine-grained criteria are dependent to each other?

The authors construct the two fine-grained preference datasets from two existing datasets. And the details are provided in the appendix. Not sure if there are other ways to construct the multi objective dataset for the purpose of the experiments?

---

> ### Author Response · Authors · 2025-11-18
> **Author Response to Reviewer KGaJ**
>
> Dear Reviewer KGaJ:
>
> Thank you for the valuable feedback on our paper. We appreciate the time and effort you have put into reviewing our work . We have carefully read your review and will respond to your questions as follows.
>
> **Q1: Are there alternative strategies to data selection? Such as LLM distillation.**
>
> A1: Our work reveals that the PD term acts as an implicit data weight in the DFPO objective. Recasting this as a data selection problem is a theoretically sound and practically efficient approach. We agree that other valuable strategies exist:
>
> 1. **Soft-weighting:** Instead of the hard-weighting of data selection, one could apply soft-weighting by using the PD term as an explicit weight in the DPO loss over the full dataset. However, this approach, which is similar to directly optimizing DFPO (§5.2), cannot save the computational cost. And the low-value samples, even when down-weighted, still consume resources and may introduce noise.
> 2. **LLM distillation**: This is a very valuable direction, for which we see at least two feasible pathways: a) **Distilling Preference Information**: A powerful teacher LLM could replace our proxy reward modeling by scoring all samples across all fine-grained aspects to compute the PD terms. b) **Distilling Preference Data**: Instead of discarding samples of low quality, a teacher LLM could be used to fix or rewrite them to be of higher quality. This would actively curate a cleaner, higher-quality dataset.
>
> We believe the distillation strategy is very promising, as it can introduce powerful external knowledge to actively improve data quality, rather than just filtering. Its main challenges lie in the high costs of teacher models and the reliance on external models. Importantly, we wish to highlight that these strategies are not mutually exclusive; in practice, data selection and distillation can be effectively combined to maximize alignment performance. We have also added this discussion to Appendix §D.
>
> **Q2: Discussion on alternative ways to define the objective for fine-grained preference alignment.**
>
> A2: In our problem, an overall ground-truth preference is decomposed into multiple, compatible fine-grained aspects (§1). This implicitly assumes these aspects are *a* priori orthogonal and equally important, as the goal is for the policy model to perform well on the whole. Therefore, at the start of our DFPO derivation, we defined the objective (Eq. 1) as optimizing the average reward.
>
> We fully agree with your view that other "starting points" exist, which would lead to deriving different final objectives and optimization problems. We briefly discuss several of these alternatives below:
>
> 1. **Optimizing a weighted reward** $\mathbb{E}[\sum_{k}w_k r_{k}(x,y)]$**:** This implies that some aspects are more important than others, which also corresponds to the point raised by other reviewers (e.g., Reviewer xH9J) about different aspects having varying priorities.
> 2. **Robust optimization** $\mathbb{E}[ \min_{k} r_{k}(x,y) ]$**:** We could define the objective as maximizing the worst-performing aspect, thereby optimizing for a model with a guaranteed performance lower-bound across all fine-grained aspects.
> 3. **Pareto optimization:** This is a more complex but purer multi-objective optimization (MOO) problem. We would not aggregate the multiple aspects into one, meaning we would not seek a single optimal model but rather a set of models representing different trade-offs. This is a completely different paradigm, which we also mentioned in §2.1 and Appendix §D.
>
> In summary, our work starts from one of the most direct and practical starting points for fine-grained preference alignment and focuses on addressing the data quality issues inherent within it. We have added this discussion to Appendix §D: Limitations and Future Work in our paper to explore these potential and diverse research directions on fine-grained preference alignment.

---

> ### Author Response · Authors · 2025-11-18
> **Author Response to Reviewer KGaJ**
>
> **Q3: What if these K fine-grained criteria are dependent to each other?**
>
> A3: Thank you for this crucial question. Our work assumes the decomposition of an overall preference into multiple, compatible aspects. We agree that dependencies can exist in some scenarios, which we categorize into two cases:
>
> 1. **Positive dependence:** E.g., "helpfulness" and "usefulness". This could over-weight similar aspects in the current PD calculation. A valuable extension is to quantify this correlation and down-weight redundant aspects in the PD estimation for a more precise consensus measure.
> 2. **Negative dependence:** E.g., "conciseness" vs. "completeness" (as also noted by Reviewer xH9J). These fine-grained criteria are definitionally in conflict, meaning a response that excels in all aspects could not exist. As discussed in our previous answer, this fundamentally changes the problem to one of Pareto optimization, which is outside our current scope of seeking a single, best-performing model.
>
> In summary, our work can be naturally extended for positive dependencies by introducing correlation weights to mitigate redundancy. The case of negative dependencies represents a different but important research direction, as further discussed in Appendix §D.
>
> **Q4: Are there other ways to construct the fine-grained preference dataset?**
>
> A4: Thank you for this practical question. First, any existing dataset providing multi-aspect preference annotations could be adapted using a similar construction pipeline mentioned in Appendix §C.1. We have also added a detailed rationale of the dataset used in Appendix §C.2.
>
> Other ways to collect such datasets are feasible and highly practical, where data could be collected as follows: 1) Define K fine-grained aspects for a task. 2) Divide the collected paired responses into K disjoint subsets. 3) Have annotators label each subset using only one specific aspect's criteria. 4) Finally, aggregate these K labeled subsets. As we further discussed in §C.2, this collection is often easier than directly annotating an overall preference (as the fine-grained criterion is simpler) and does not increase the annotation burden (each sample is still labeled only once). Following this preliminary pipeline, we have also collected a fine-grained dataset for our proprietary real-world application to validate the effectiveness of our method. Details and Results have been added in §5.7 and Appendix §C.3.

---

### Author Response · Authors · 2025-12-02
**Author Summary of Rebuttal to Area Chair**

Dear Area Chair,

We sincerely appreciate your time and effort in handling our submission. We fully understand the significant workload involved due to the system rollback; to facilitate your efficient assessment, we provide this concise summary of our rebuttal progress.

Before the system rollback, we had actively engaged with all reviewers to address their questions and concerns in our rebuttal and manuscript revision. During the valid rebuttal period, one reviewer (xH9J) explicitly acknowledged our responses and further raised the score. Unfortunately, the other three reviewers were unable to provide final feedback in time, likely due to the system disruption.

Below is a concise summary of how we responded to the key points from each reviewer and the corresponding improvements made to the paper.

**1. Response to Reviewers**

- **Reviewer KGaJ** (original score: 6) assessed the paper positively while raising constructive exploratory questions regarding alternative optimization objectives, integration with LLM distillation, dependencies between criteria, and practical dataset construction. We fully addressed these points by providing comprehensive clarifications and significantly expanding Appendix D to include detailed discussions on these future directions and practical implementations.
- **Reviewer vgsY** (original score: 4) acknowledged our clear motivation, rigorous analysis, and extensive evaluation. They raised some questions about the benchmark used and hyperparameter sensitivity. As the reviewer candidly noted, "I am not familiar with this research scope", we first specifically clarified the rationale for our dataset choice, and then added a completely new experiment on a real-world application (§5.7) to support our claim. Furthermore, we added comprehensive ablation studies (§5.6) on hyperparameters to demonstrate the method's robustness.
- **Reviewer qeXk** (original score: 8) strongly supported the paper. The reviewer raised insightful questions and discussions regarding the urgency of the problem in the era of scaling laws, the mechanism of handling definitionally conflicting preferences, the ground-truth preference in multi-task settings, and potential extensions to iterative regimes. We addressed all points by citing recent literature to demonstrate that data selection remains critical in post-training, clarifying the difference between annotation quality issues and incompatible objectives, and expanding Appendix D to discuss iterative extension and future Pareto optimization directions. Additionally, as kindly suggested, we have enriched the analysis of our experiments in the revised manuscript.
- **Reviewer xH9J** (original score: 6 $\rightarrow$ 8) acknowledged the progression of our motivation and method, noting the experiments were thorough and convincing. They raised constructive questions regarding the proxy reward models, hyperparameter sensitivity, aspect correlations, and the equal weighting assumption. We addressed these by providing detailed clarifications and adding new ablation studies (§5.6) and expanding Appendix D to discuss weighted objectives and iterative training. Additionally, as kindly suggested, we added further explanations to the mathematical derivation to improve comprehension. The reviewer actively engaged with our response and raised their score before the system incident.

**2. Summary of Paper Revisions**

During the rebuttal period, we made improvements to the manuscript (highlighted in blue). Key revisions include:

- **New real-world experiment** (§5.7): We added a completely new experiment on a real-world downstream application to demonstrate the method's effectiveness (addressing Reviewer vgsY).
- **New ablation studies** (§5.6): We added detailed sensitivity analyses for key hyperparameters, confirming the method's stability (addressing Reviewers vgsY & xH9J).
- **Expanded discussion** (Appendix D): We significantly broadened the discussion on limitations and future directions, specifically covering reward modeling challenges, dataset availability, iterative extensions, alternative optimization objectives, and integration with other techniques (addressing Reviewers KGaJ & qeXk).
- **Enhanced analysis and derivation** (§5 & Appendix A.1.1): We improved the clarity of the mathematical derivation and enriched the analysis of experimental results to provide deeper insights (addressing Reviewers xH9J & qeXk).

Finally, we extend our sincere gratitude to all reviewers for their dedicated time and constructive feedback. We have actively engaged in the discussions and incorporated their suggestions to enhance the quality of our manuscript. We hope this summary provides a clear overview of our efforts and assists you in your final assessment.

Thank you again for your time and service to the community.

Sincerely,

The Authors

---

### Meta-Review · Area_Chair_nbiL · 2026-01-10

**Summary:**

This paper proposes a data-centric approach for Large Language Model (LLM) alignment by leveraging fine-grained, aspect-specific preferences. The authors introduce *Preference Divergence (PD)* to quantify conflicts between different preference aspects in aggregated datasets. Rather than directly optimizing a complex objective, they recast the problem as a data selection task. Their strategy identifies a subset of data with the most negative PD values, which represent instances where fine-grained preferences are most consistent, for efficient training. Theoretical analysis and empirical results demonstrate that this method can outperform standard full-data alignment while using as little as 30% of the dataset.

Overall, this paper gives a nice contribution to the problem of dataset conflicts. In particular, the paper has the following favorable properties:
- Practical and Novel Algorithm: The proposed data selection algorithm is well-defined, executable, and exhibits clear novelty in the field of LLM alignment.
- Clarity of the method: The derivation of the selection strategy is explained clearly and provides a high level of conviction regarding its mathematical soundness. The framework effectively addresses the issue of data quality. It focuses on removing samples that appear conflicted due to noise or poor labeling rather than trying to reconcile fundamentally irreconcilable human values, making the target of the optimization clear and effective.

On the other hand, this paper can be improved in the following aspects:
- Notation Clarity: The current notation needs refinement for better readability. Specifically, the notation for the model $M_\theta$ should be improved to clearly indicate its dependency on the specific aspect $k$.
- Selection of $k$: The criteria and rationale for how a specific aspect $k$ is chosen within the framework were somewhat ambiguous and could benefit from a more detailed explanation in the final version.
- Data Efficiency Claims: While the results at 30% data usage are impressive, further discussion on the limits of this efficiency across even more diverse datasets would be valuable.

Despite (rather) minor issues as listed above, the paper presents a solid, and practically useful contribution to LLM alignment. The data-centric perspective on resolving preference conflicts through PD-based selection is a valuable addition to the community. Therefore, this paper can be accepted to ICLR.

**Reviewer Concerns:**

I did not find any concern on the reviews.

**Reviewer Scores:**

The score of Reviewer vgsY could be updated to 6 from 4 because their concerns were mostly addressed and their confident is not high.

---

### Decision · Program_Chairs · 2026-01-26

Accept (Poster)